# Graph Coloring via Locally-Active Memristor Oscillatory Networks

Alon Ascoli [1,*] , Martin Weiher [1], Melanie Herzig [2], Stefan Slesazeck [2], Thomas Mikolajick [2,3]
and Ronald Tetzlaff [1]

1   Chair of Fundamentals of Electrical Engineering, Institute of Circuits and Systems, Faculty of Electrical and Computer Engineering, Technische Universität Dresden, 01062 Dresden, Germany; martin.weiher@tu-dresden.de (M.W.); ronald.tetzlaff@tu-dresden.de (R.T.)
2   Nano-Electronic Materials Laboratory (NaMLab) gGmbH, 01187 Dresden, Germany; melanie.herzig@namlab.com (M.H.); stefan.slesazeck@namlab.com (S.S.); thomas.mikolajick@namlab.com (T.M.)
3   Institute für Halbleiter-und Mikrosystemtechnik, Technische Universität Dresden, 01062 Dresden, Germany
*   Correspondence: alon.ascoli@tu-dresden.de

**Abstract:** This manuscript provides a comprehensive tutorial on the operating principles of a bio-inspired Cellular Nonlinear Network, leveraging the local activity of $NbO_x$ memristors to apply a spike-based computing paradigm, which is expected to deliver such a separation between the steady-state phases of its capacitively-coupled oscillators, relative to a reference cell, as to unveil the classification of the nodes of the associated graphs into the least number of groups, according to the rules of a non-deterministic polynomial-hard combinatorial optimization problem, known as vertex coloring. Besides providing the theoretical foundations of the bio-inspired signal-processing paradigm, implemented by the proposed Memristor Oscillatory Network, and presenting pedagogical examples, illustrating how the phase dynamics of the memristive computing engine enables to solve the graph coloring problem, the paper further presents strategies to compensate for an imbalance in the number of couplings per oscillator, to counteract the intrinsic variability observed in the electrical behaviours of memristor samples from the same batch, and to prevent the impasse appearing when the array attains a steady-state corresponding to a local minimum of the optimization goal. The proposed Memristor Cellular Nonlinear Network, endowed with ad hoc circuitry for the implementation of these control strategies, is found to classify the vertices of a wide set of graphs in a number of color groups lower than the cardinality of the set of colors identified by traditional either software or hardware competitor systems. Given that, under nominal operating conditions, a biological system, such as the brain, is naturally capable to optimise energy consumption in problem-solving activities, the capability of locally-active memristor nanotechnologies to enable the circuit implementation of bio-inspired signal processing paradigms is expected to pave the way toward electronics with higher time and energy efficiency than state-of-the-art purely-CMOS hardware.

**Keywords:** graph coloring; cellular nonlinear networks; memristor oscillatory networks; locally-active memristors; control theory

## 1. Introduction

Memristor technologies promise to revolutionise the world of electronics in the years to come, allowing to boost the performance of integrated circuits beyond the Moore era. Theoretically introduced in 1971 by L. Chua [1], memristors are essentially resistances with state- and input-dependent programmable capability [2]. Despite the first association between Chua's theory and the experimental observation of fingerprints of memristive behaviour at the nanoscale was made by R.S. Williams and his team at Hewlett Packard Labs in 2008 [3], the appearance of memory resistance switching effects in miniaturized physical structures were reported in numerous occasions throughout the past two centuries [4],

constituting the object of extensive and intensive investigations for the development of novel solid-state memories first in the 1960s [5]. In fact, the main application of memristors [6]—certainly the most profitable one from a business perspective point of view—is the design of memories with higher retention, lower power consumption, and larger density as compared to state-of-the-art data storage units [7]. Another major field of application regards the development of innovative neuromorphic systems, which resemble biological entities more closely than traditional artificial networks ([8–11]). A closely-related branch of research takes inspiration from the high levels of organization and energy efficiency of biological systems to develop mem-computing machines ([12–14]), which extend the functionalities of traditional cellular architectures [15]. Another bio-inspired research direction aims to the circuit implementation of in-memory computing paradigms through high-capacity memristive memories, stacked in 3D crossbar arrangement above underlying CMOS circuitry [16], and employed, alternatively, to store data or to execute processing tasks ([17–20]), which reveals the high potential of novel hardware platforms of this kind to resolve the von Neumann bottleneck, limiting the maximum operating speed of traditional computing engines, in the near future. Additionally, the unique combined capability of non-volatile resistance switching memories to sense data ([21,22]), learn how to recognize patterns [23], process information [24], and store multiple states [25] within a single tiny physical volume, and the availability of nanoscale locally-active [26] volatile memristors, which may amplify the small signal upon suitable polarization ([27,28]), open up yet-unexplored opportunities for the Internet-of-Things (IoT) industry, which urgently calls for the development of miniaturized, low-power, light-weight, portable, and smart technical systems, which, within a very short time frame, are able to acquire a large amount of information from the environment, to extract features of interest from noisy data so as to solve specific optimization problems, and to store or transmit to a prescribed user the most relevant results of the computation.

One of the most challenging tasks for computing machines, based upon the classical von Neumann architecture, is the solution of combinatorial optimisation problems belonging to the non-deterministic polynomial (NP)-hard complexity class[1]. Nowadays there is a huge interest in developing novel inexpensive low-power high-speed hardware solutions capable to solve NP-hard problems more efficiently than traditional computers, given that high-performance technical systems of this kind may find application in various industry sectors, e.g., for traffic management, airline scheduling, gene sequencing, and electronic chip wiring.

A recent Nature Electronics publication [29] proposed the use of a memristive crossbar array—refer to Figure 1—for accelerating the vector-matrix multiplications (VMMs) at the basis of the update rule of an iterative machine learning algorithm, which is credited to Hopfield, and enables to solve combinatorial problems, within an overall disruptive analogue hardware architecture, leveraging and controlling its numerous inherent noise sources to allow a power-efficient derivation of the optimal solutions.

---

[1] The time it takes for a von Neumann computing machine to find the optimal solution to a NP-hard problem, which involves $n$ elements, scales exponentially with n.

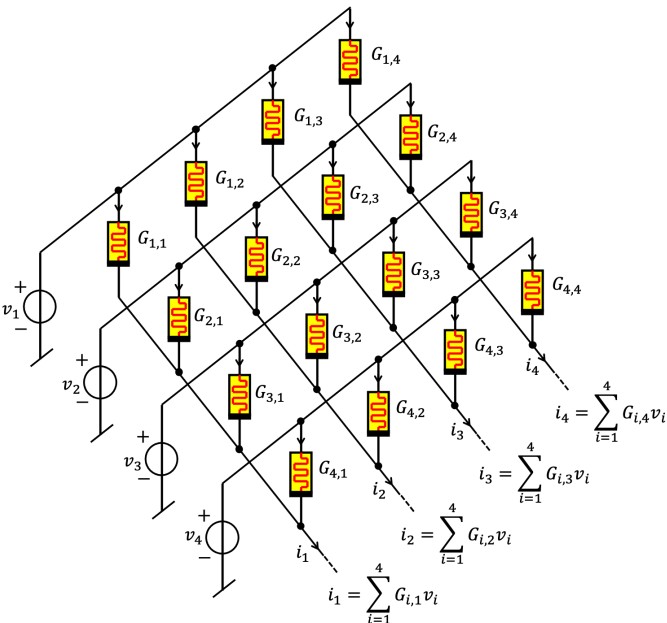

**Figure 1.** In-memory computing in a $N \times N$ memristive crossbar array (here $N = 4$). In-memory computing in an $N \times N$ memristive crossbar array (here, $N = 4$). Naturally obeying Kirchhoff's Current Law (KCL), the bio-inspired network enables a time-efficient computation of VMMs. With $j \in \{1, \ldots, N\}$, the current flowing down the $j^{th}$ column of the array is simply given by $i_j = \sum_{i=1}^{N} G_{i,j} \cdot v_i$, where $G_{i,j}$ denotes the conductance of the memory resistive switch located at the intersection between the conductive nanowires stretching along row $i$ and column $j$ [29]. The computation of the currents at the outputs of the crossbar columns assumes that the bottom terminals of all the memristors—refer to the thick black horizontal segments in their circuit-theoretic symbols—are at virtual ground.

Cellular Nonlinear Networks (CNNs) with locally-active volatile memristors [30] constitute a powerful engine for the implementation of bio-inspired spike-based computing paradigms. This manuscript, inspired to a recent publication [31], is devoted to explain in a pedagogical form how these bio-inspired memristor cellular arrays[2], may be adopted for solving a complex NP-hard problem known as vertex coloring. While page limitation prevented a complete description of the theory at the basis of the spike-based computing paradigm implemented by the proposed cellular arrays[3], all details are pedagogically reported in this tutorial. In regard to the structure of the manuscript, Section 2 provides a brief description of the memristor model adopted for the study. Section 3 introduces the memristive computing engine for solving the vertex coloring problem, including a discussion of its operating principles, and the specification of strategies aimed to compensate for non-idealities, including an imbalance in the number of couplings per oscillator, and the memristor device-to-device variability. Section 4 presents a rigorous iterative procedure for coloring a graph via the network phase dynamics. Importantly, control paradigms to resolve local minima-based impasse conditions [33] are proposed in Section 5, which further compares the performance of the proposed spike-based computing engine, endowed with ad hoc circuitry to implement such strategies, with the solutions of state-of-the-art software and hardware competitor systems. Finally, conclusions are drawn in Section 6.

---

2    CNNs with non-volatile memristors ([12–14]) may pave the way toward the development of advanced visual-sensor processors [32] with high spatial resolution and intrinsic memory capability.

3    Importantly, depending upon the graph under focus, the proposed Memristor CNN (M-CNN) may feature either local or non-local capacitive couplings. The solution of the vertex coloring problem through the proposed M-CNNs depends upon the phase differences among the oscillations developing in the constitutive units of the array at steady state. In order to highlight the steady-state oscillatory behaviours of the cells during operation, the bio-inspired arrays are also referred to as Memristor Oscillatory Networks (MONs) in the remainder of the manuscript.

## 2. A Physics-Based Model for the Threshold Switching Dynamics of a Nano-Scale Locally-Active Memristor Device Stack

In a past study [34,35] we employed physics laws to construct state evolution and memductance functions of a NbO$_x$ memristor, fabricated at NaMLab, after inferring the physical mechanisms, which underlie its nonlinear dynamics, from the outcome of experimental measurements, and the insights gained through theoretic investigations of a mathematical model, derived previously on the basis of Chua's Unfolding Theorem [28]. A thorough analysis of the proposed physics-based model [34] revealed that *the Mott insulator-to-metal transition does not constitute the key physical mechanism at the origin of the threshold switching process, that each of our NbO$_x$-based memristors undergoes under the application of a generic quasi-static voltage stimulus between its two terminals.* Conversely, a temperature-activated trap-assisted Poole-Frenkel conduction mechanism underlies the abrupt turn-on dynamics of the volatile memristor. Importantly, shortly after our discovery, a further proof of evidence for the validity of our conjecture was provided by engineers from Hewlett Packard (HP) Labs [36]. Remarkably, despite it was originally proposed for the NbO microstructure, the physics-based model in [34] was found to fit rather well also experimental data extracted from nanoscale variants of the NbO$_x$ threshold switching resistance from NaMLab after minor adaptations [37,38].

As illustrated in Figure 2(a), ([30,31]), the equivalent circuit model of the memristor $\mathcal{M}$ consists of the series combination between a linear resistor $R_c$, capturing the action of the top electrode resistance, and a parallel one-port, formed by a core memristor $\tilde{\mathcal{M}}$, and a nonlinear resistor $\mathcal{R}$, which accounts for the parasitics inherent to the NbO$_x$ nanostructure, and is responsible for the manifestation of leakage current effects. Several studies [34,35] have revealed that the state of the core memristor is well captured by its body temperature $T$. In fact, as anticipated earlier, threshold switching effects in the nano-device originate from runaway Joule self-heating governed by Poole-Frenkel electrical conduction mechanisms. Taking this into account for the formulation of a state-dependent Ohm's law, with $\tilde{v}_m$ ($\tilde{i}_m$) representing the voltage (current) of $\tilde{\mathcal{M}}$, and choosing Newton's law of cooling to dictate the time evolution of the state, the DAE set, governing the static and dynamic behaviour of the core memristor may be expressed as

$$\frac{dT}{dt} = g(T, \tilde{v}_m) \triangleq \frac{1}{C_{th}} \cdot \tilde{i}_m \cdot \tilde{v}_m - \frac{\Gamma_{th}}{C_{th}} \cdot (T - T_{amb}), \tag{1}$$

$$\tilde{i}_m = G(T, \tilde{v}_m) \cdot \tilde{v}_m \triangleq \frac{1}{R_{01}} \cdot \exp\left(-\frac{a_{01} - a_{11} \cdot |\tilde{v}_m|}{T}\right) \cdot \tilde{v}_m, \tag{2}$$

where $C_{th}$ ($\Gamma_{th}$) stands for the effective thermal capacitance (conductance) of the core device $\tilde{\mathcal{M}}$, $T_{amb}$ denotes the ambient temperature, $R_{01}$, $a_{01}$ and $a_{11}$ are constants[4], while $v_m$ ($i_m$) symbolises the voltage (current) falling across (flowing though) the memristor $\mathcal{M}$, whose circuit-theoretic symbol is shown in Figure 2(b). Remarkably, the DAE set (1)–(2) of the core memristor falls into the voltage-controlled extended memristor family from Chua's classification[5] [40]. The constitutive relationship $f(v_{\mathcal{R}}, i_{\mathcal{R}}) = 0$ of the nonlinear resistor $\mathcal{R}$ is given in implicit form as

$$v_{\mathcal{R}} = R_{02} \cdot i_{\mathcal{R}} \cdot \exp\left(\frac{a_{02} - a_{12} \cdot \sqrt{|v_{\mathcal{R}}|}}{T_{amb}}\right). \tag{3}$$

---

[4]  The reader is invited to consult [34,35] for details on the association between the real parameters $R_{01}$, $a_{01}$ and $a_{11}$ and the physical properties of the core nanostructure.

[5]  It is instructive to observe that there exist memristor physical realisations, whose models feature an even more general input- and state- dependent Ohm law than what is admissible for extended memristors. For these two-terminal devices, including the TiO$_2$ memristor from HP Labs [39], the mathematical description includes an implicit Ohm law of the form $h(x, v_m, i_m) = 0$, with $x$, $v_m$, and $i_m$ denoting the device state, voltage, and current, respectively.

revealing that a variant of the Poole-Frenkel law explains current transport phenomena in the parasitic resistor as well[6]. The nominal parameter values for the core memristor and nonlinear resistor models were obtained by fitting the underlying equations to experimental data extracted from nano-device samples (see [37,38] for details on the $NbO_x$ nanostructure fabrication process). Importantly, since the non-negligible intrinsic spread in dynamic behaviour from sample to sample may impair the capability of a computing engine based on memristive hardware to perform a predefined data processing task as desired, this non-ideality may not be neglected in circuit design considerations. Particularly, in the analysis to follow, where a Memristor Oscillatory Network (MON) is adopted to find optimal solutions to graph coloring problems [31], it will be accounted through the replacement of specific parameters in the $NbO_x$ nano-scale threshold switch physics model, specifically $\Gamma_{th}$, $R_{01}$, $a_{01}$, $a_{11}$, $R_C$, $R_{02}$, and $a_{12}$, with corresponding ones, i.e., in turn, $\Gamma_{th,\alpha}$, $R_{01,\alpha}$, $a_{01,\alpha}$, $a_{11,\alpha}$, $R_{C,\alpha}$, $R_{02,\alpha}$, and $a_{12,\alpha}$, that are controlled via a real variable $\alpha$, which is set randomly to a distinct value chosen from a uniform distribution across the closed range $[0, 1]$ for each nanostructure individually, prior that the simulation of the ODE, modelling the array associated to a pre-specified graph, is commenced.

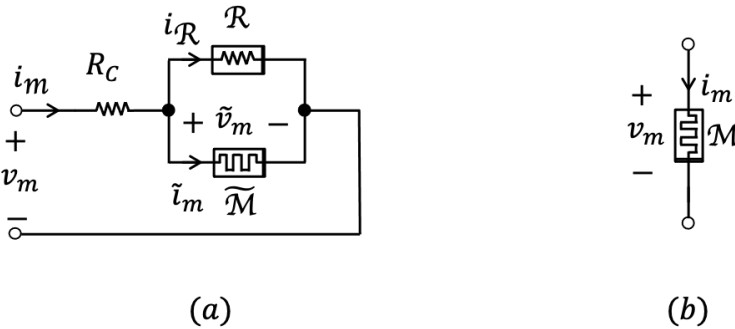

**Figure 2.** (**a**) Equivalent circuit of the physical model of a $NbO_x$ nanoscale memristor $\mathcal{M}$ from NaMLab. The linear resistor $R_C$ and the nonlinear resistor $\mathcal{R}$ respectively account for the effects of electrode contact resistance and parasitics. (**b**) Memristor circuit-theoretic symbol.

Table 1 reports the parameter setting of the nano-scale memristor physics-based model modulated according to the device-to-device variability estimated statistically beforehand through the analysis of current-voltage characteristics of a large number of samples under a common quasi-static stimulation [31].

**Table 1.** Parameter setting for $NbO_x$ nanoscale threshold switch from NaMLab. The effects of the memristor-to-memristor variability are accounted through the assignment of a distinct value, chosen randomly within the closed set $[0, 1]$ to the variable $\alpha$, controlling specific coefficients of Equations (1), (2), and (3).

| $C_{th}$ / J $\cdot$ K$^{-1}$ | $\Gamma_{th,\alpha}$ / W $\cdot$ K$^{-1}$ | $T_{amb}$ / K | $R_{01,\alpha}$ / $\Omega$ | $a_{01,\alpha}$ / K |
|---|---|---|---|---|
| $1 \cdot 10^{-14}$ | $1.889 \cdot 10^{-6} \cdot 1.064^{\alpha}$ | 293 | $3.047 \cdot 0.831^{\alpha}$ | $3620 \cdot 1.061^{\alpha}$ |

| $a_{11,\alpha}$ / K $\cdot$ V$^{-1}$ | $R_{c,\alpha}$ / $\Omega$ | $R_{02,\alpha}$ / $\Omega$ | $a_{02}$ / K | $a_{12,\alpha}$ / K $\cdot$ V$^{-1/2}$ |
|---|---|---|---|---|
| $820.4 \cdot 1.137^{\alpha}$ | $173.8 \cdot 1.092^{\alpha}$ | $565 \cdot 1.377^{\alpha}$ | 1000 | $168.8 \cdot 1.083^{\alpha}$ |

## 3. Memristive Computing Engine for Solving the Vertex Coloring Problem

One of the most popular NP-hard problems is *graph or vertex coloring*. Given an undirected graph, consisting of a certain arrangement of edge-coupled vertices, the aim of the problem is to assign a color to each vertex, satisfying the constraint, which dictates

---

[6] The mathematical description of the nonlinear resistor, formulated in Equation (3), is in fact equivalent to the model of a $NbO_x$ memristor, as originally presented in [34,35], which reveals the correspondence of the real parameters $R_{02}$, $a_{02}$ and $a_{12}$ to physical properties of the nanostructure, in the limit when changes occurring in the device state, defined as its body temperature, are negligible.

that a given color should be shared between as many vertices as possible so long as no edge connects any two of them. The lowest number of color groups, the vertices of the unconnected graph may be classified into, is called *chromatic number*.

As revealed back in 1988, graph coloring may be achieved by harnessing synchronisation mechanisms in arrays of of coupled oscillatory cells [41]. A more recent work [42] showed that the assignment of colors to vertices of a graph may be naturally implemented through the analysis of the steady-state phase shifts between capacitively-coupled relaxation oscillators exploiting negative differential resistance (NDR) effects in vanadium dioxide ($VO_2$) nano-structures. Inspired from this research, we have recently investigated the capability of an oscillatory network, leveraging locally-active dynamics in $NbO_x$ memristors, and featuring capacitive couplings, to identify the minimum possible number of colors assignable to the vertices of an associated undirected graph via phase dynamics.

In order to color a given graph of N vertices or nodes, a unique number in the set $\{0, 1, \dots, N-1\}$ is first attributed to each vertex. A one-to-one association is then established between the vertices (edges) of the graph, and the oscillators (coupling capacitors, each of capacitance $C_C$) of the associated network. The oscillator 0, corresponding to the node 0, assumes a critical role in the solution of the graph coloring problem, and is called *reference cell*. A general indication, regarding the selection of a suitable reference oscillator among the N possible candidates, will be given shortly. As an example, Figure 3(a) and (b) show a 6-node ring and the associated MON, respectively.

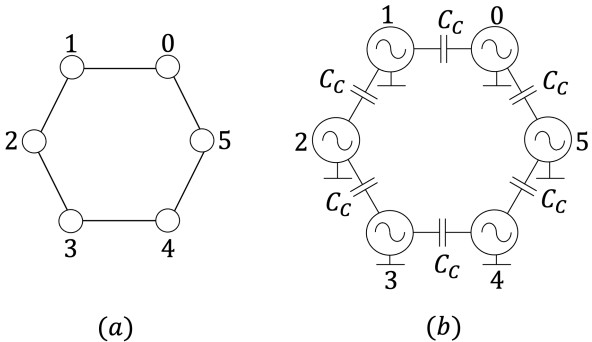

**Figure 3.** (**a**) A 6-node ring-shaped undirected graph (**b**) Associated MON. The oscillator *i* of the network corresponds to the vertex *i* of the graph ($i \in \{0, 1, 2, 3, 4, 5\}$).

With reference to Figure 4, plots (a) and (b) show the oscillator circuit and its symbol, respectively. Each oscillatory cell is composed of the parallel connection between a $NbO_x$ memristor $\mathcal{M}$, a bias circuit, consisting of the series combination of a DC voltage source $V_S$ with a series resistor $R_S$, allowing to polarize [28] the resistance switching memory within the locally-active region of its DC current-voltage locus, and a capacitor $C$.

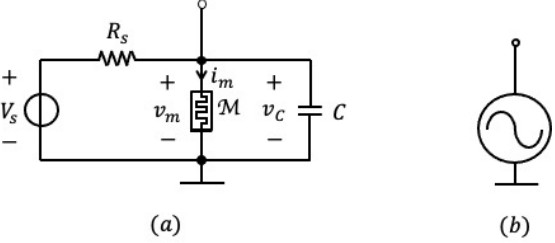

**Figure 4.** Memristive oscillatory cell (**a**) and its symbol (**b**).

On the basis of the $NbO_x$ nano-device physics-based model, expressed by Equations (1), (2), and (3), under the variability-aware parameter setting of Table 1, we carried out a deep numerical investigation of the capability of an array of capacitively-coupled memristive

oscillatory cells to classify the vertices of a pre-defined undirected graph in the least number of groups.

While the memristor model parameters, accounting for the inherent fluctuations in static and dynamic properties among distinct nano-device samples, were already reported in Table 1, only the values assigned to the physical quantities of the non-memristive circuit elements in the proposed MON are provided in Table 2.

**Table 2.** Parameter setting for the non-memristive circuit elements in the oscillator of Figure 4(a).

| $V_S$/ V | $R_S$/ $\Omega$ | $C$/ F | $C_C$/ F |
|---|---|---|---|
| 2.5 | 5525 | $10 \cdot 10^{-9}$ | $0.2 \cdot 10^{-9}$ |

### 3.1. Operating Principles of the Capacitively-Coupled Networks

The capacitive nature of the couplings in the network is responsible for pulling the phases of physically-connected oscillators far apart one from the other at steady-state. This repelling mechanism may be exploited to colour the vertices of the associated graph. The phases of uncoupled oscillators tend to form clusters, which may be interpreted as color groups for the corresponding vertices. The larger is the separation between phase clusters, the simpler is the classification of the nodes of the graph into color groups.

To illustrate this concept, let us consider a simple undirected graph, composed of one edge, which couples two nodes, as shown in Figure 5(a). The chromatic number of this graph is obviously equal to 2. The corresponding oscillatory network is depicted in plot (b) of the same figure. Simulating the circuit with nominal parameter setting[7], the time waveforms of the voltages across the capacitors or those of the currents through the memristors within the circuits of the two capacitively-coupled memristive oscillators are expected to feature a steady-state phase shift of about 180°. Upon the emergence of anti-phase synchronisation between the two oscillators of this simple network, it would be natural to assign one color to vertex 0 and another one to vertex 1 of the associated graph of Figure 5(a).

As a further example, Figure 5(c) visualises another undirected graph with chromatic number equal to 2. The respective oscillatory array is shown in plot (d) of the same figure. Simulating this network, the phases of oscillators 1 and 2 are expected to cluster together, and to shift away from the phase of reference oscillator 0 as much as possible, approaching a relative value of approximately 180° at steady state. With the network exhibiting such a phase pattern at steady state, it would be straightforward to divide the vertices of the associated graph into two color groups, including vertex 0 and vertices 1 and 2, respectively.

Due to a couple of non-idealities the expectations on the phase dynamics of the networks in plots (b) and (d) of Figure 5 are not fulfilled in practice. Details will be provided in the next two sections.

---

[7]　The nominal parameter setting is obtained from Table 1 for $\alpha = 0.5$. As will be shown later, simulating the memristor model under a quasi-DC voltage stimulus and with the variability parameter stepped across its existence domain, the locus observed for $\alpha = 0.5$ appears in the center of the distribution of characteristics emerging in the voltage-current plane.

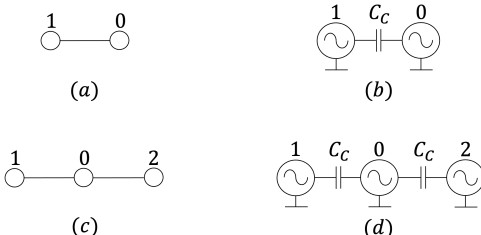

**Figure 5.** (**a**) A 2-node 1-edge graph. Its chromatic number is 2. (**b**) Oscillatory network corresponding to the graph in (**a**). (**c**) A 3-vertex 2-edge graph. Its chromatic number is once again 2. Interestingly, the number of edges departing from vertex 0 (from either vertex 1 or vertex 2) is 2 (1). (**d**) Oscillatory network corresponding to the graph in (**c**).

Before proceeding, the following remark explains how the autonomous memristive array is initialised, and clarifies how the steady-state phases of the cells are computed when the network converges to an oscillatory solution.

**Remark 1.** *With $V_S$ and $R_S$ fixed to specific values, as reported in Table 1, all memristors in the network are biased in a common operating point lying along the NDR of the DC $I_m$–$V_m$ locus of the NbO$_x$ resistive nanoswitch. Defining as $v_{C,i}$ and $T_i$ the states of the second-order cell i of the memristive array ($i \in \{0, \ldots, N-1\}$), the initial conditions for the capacitor voltage and the memristor[8] temperature are set in each oscillator to $0\,V$, and to the ambient temperature $T_{amb}$, fixed to $293\,K$ in Table 1, respectively. A random sequence is generated to mismatch, in a non-deterministic way, the time instants, at which the signals generated by the DC voltage sources within the oscillators ramp toward the nominal $V_S$ value over a time span of $1\,\mu s$ at the beginning of a simulation. If sustained periodic oscillations develop across the network at steady state, for each oscillator $i \in \{0, \ldots, N-1\}$, the phase of the memristor current $i_{m,i}$ relative to the phase of the current $i_{m,0}$ through the NbO$_x$ device in the reference oscillator 0 is then computed, as follows. First, the common period T of the oscillations, observed in the time waveforms of the memristor currents at steady state, is estimated. For each i-value in the set $\{0, \ldots, N-1\}$, the time instant $t_i$, at which the memristor current $i_{m,i}$ in the cell i attains a given threshold value $I_{th}$, specifically $0.5\,mA$, during its ascending phase, within a single common steady-state cycle, is then recorded. The cycle, utilised for these calculations, covers the time span $[t_0, t_0 + T]$, where $t_0$ marks the time instant, when this threshold crossing event occurs for the memristor current $i_{m,0}$ in the reference cell 0. Next, for each i-value, the temporal span $\Delta t_i \triangleq t_i - t_0$, which separates the instants $t_i$ and $t_0$, at which the threshold crossing event occurs for the memristor currents in the cells i and 0, respectively, is calculated. Finally, this allows to compute the steady-state phase shift between cells i and 0 via[9] $\varphi_i^{(s)} \triangleq \omega_0 \cdot \Delta t_i$, where $\omega_0 \triangleq \frac{2\pi}{T}$, for each i-value.*

### 3.2. Compensation for an Imbalance in the Number of Couplings per Oscillator

If an imbalance in the number of edges per node characterises the coupling structure of a given graph, the phases of physically-coupled oscillators in the corresponding memristive oscillatory network may be found to hold only a marginal distance one from the other at steady state. The balanced nature of the graph of Figure 3 (Figure 5(a)) originates from the fact that each of its six (two) nodes is coupled to 2 other nodes (the other node). On the other hand, inspecting the graph of Figure 5(c), vertex 0 is coupled to vertices 1 and 2, while each of vertices 1 and 2 is connected to vertex 0 only. The resulting imbalance in the number of couplings per cell, arising in the associated array of memristive oscillators—refer to Figure 5(d)—prevents the phases of cells 1 and 2 from separating as expected from the

---

[8]    The voltage across (current through) the memristor in the cell i is indicated via $v_{m,i}$ ($i_{m,i}$).

[9]    The units of the steady-state relative phase of oscillator i, computed via $\varphi_i^{(s)} = \omega_0 \cdot \Delta t_i$, are radiants. In order to express $\varphi_i^{(s)}$ in degrees, its formula needs to be scaled by the factor $\frac{180°}{\pi}$ ($i \in \{0, \ldots, N-1\}$).

phase of cell 0. This is shown in the phase diagram[10] of Figure 6(a), where the dashed orange and green traces, illustrating the time evolution of the phases of cells 1 and 2 relative to cell 0, respectively, feature a separation as small as 50° from the reference 0° level at steady state. The unbalanced nature of the network of Figure 5(d) results in an imbalance in the capacitive load per oscillator. Looking at the coupling configuration in this array, and recalling the circuit of each oscillator, shown in Figure 4(a), in which, for simplicity, the memristor is replaced with its small-signal equivalent circuit model [43], the application of basic circuit-theoretic principles allows to obtain an expression in the sinusoidal regime for the capacitive impedance $Z_{C_i}(j\omega)$, loading oscillator $i$ for each value of $i$ in the set $\{0, 1, 2\}$, i.e.,

$$
\begin{align}
Z_{C_0}(j\omega) &= Z_C \parallel (Z_{C_C}(j\omega) + Z_C(j\omega)) \parallel (Z_{C_C}(j\omega) + Z_C(j\omega)), \tag{4}\\
Z_{C_1}(j\omega) &= Z_C \parallel \left\{ Z_{C_C}(j\omega) + \left[ Z_C(j\omega) \parallel (Z_{C_C}(j\omega) + Z_C(j\omega)) \right] \right\}, \tag{5}\\
Z_{C_2}(j\omega) &= Z_C \parallel \left\{ Z_{C_C}(j\omega) + \left[ Z_C(j\omega) \parallel (Z_{C_C}(j\omega) + Z_C(j\omega)) \right] \right\}, \tag{6}
\end{align}
$$

where

$$
Z_C(j\omega) = \frac{1}{j\omega C}, \quad \text{and} \tag{7}
$$

$$
Z_{C_C}(j\omega) = \frac{1}{j\omega C_C}. \tag{8}
$$

Defining

$$
C_a \parallel C_b \triangleq \frac{C_a \cdot C_b}{C_a + C_b}, \tag{9}
$$

the load capacitance $C_i$ of the oscillator $i \in \{0, 1, 2\}$ in the network of Figure 5(d) may be extracted easily from the $(i+1)^{\text{th}}$ equation in the triplet (4)–(6), yielding

$$
\begin{align}
C_0 &= C + 2 \cdot (C_C \parallel C), \tag{10}\\
C_1 &= C + C_C \parallel (C + C_C \parallel C) \approx C + C_C \parallel C, \tag{11}\\
C_2 &= C + C_C \parallel (C + C_C \parallel C) \approx C + C_C \parallel C, \tag{12}
\end{align}
$$

where the approximations stem from the inequality $C \gg C_C$, yielding $C + C_C \parallel C \approx C$. Analysing Equations (10)–(12) it is clear that, in order to compensate for the imbalance in the capacitive load per oscillator within the network of Figure 5(d), an additional capacitor with capacitance $C_{\text{comp},j} = C_C \parallel C$ should be added in parallel to the capacitor $C$ in the circuit of oscillator $j$, for each $j$-value in the set $\{1, 2\}$, as shown in Figure 6(b). Simulating the balanced network, the relative phases of cells 1 and 2 cluster together, converging to values close to the expected 180° level at steady state, as may be evinced from plot (a) in the same figure, where the orange and green solid traces reveal the time evolution of $\varphi_1$ and $\varphi_2$, respectively.

---

10   A new graphical tool—which we call *phase diagram*—is introduced in this research study [31] for visualising the phase dynamics of the network. Referring, for example, to the phase diagram of Figure 6(a), a specific trace visualises the time evolution of the phase of oscillator $j$ relative to oscillator 0 ($j \in \{1, \ldots, N-1\}$). Reading the time flow along the radial direction, the angle between the segment, joining the origin to the point, where the trace is found to lie at time $t$, and the blue horizontal line, denoting the 0°-valued reference level, represents the phase shift $\varphi_j(t)$ of oscillator $j$ with respect to oscillator 0 at time $t$.

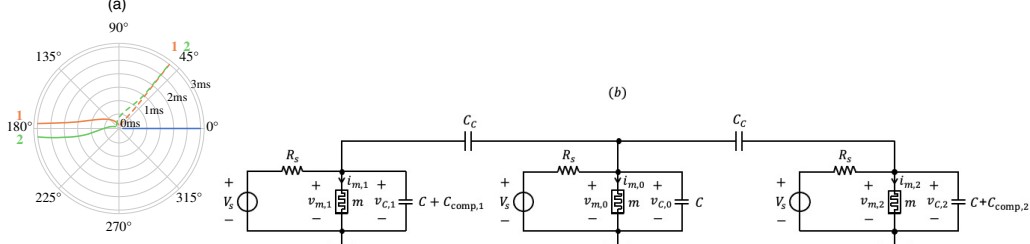

**Figure 6.** (**a**) Phase diagram visualising the time evolution of the phases of oscillators 1 (in orange) and 2 (in green) relative to oscillator 0, sitting on the 0° phase state throughout the simulation (blue horizontal line) for the original unbalanced network of Figure 5(c) (see the dashed traces) and for the compensated network in plot (**b**) of this figure (refer to the solid traces). In the first (latter) case, the phases of oscillators 1 and 2 are found to cluster together, and to distance themselves from the reference 0° phase, associated to oscillator 0, by approximately 50° (180°) at steady state. (**b**) Complete circuitry of the memristive oscillatory network of Figure 5(d) after compensation for the imbalance in the number of couplings per oscillator. Here $C_{\text{comp},1} = C_{\text{comp},2} = C_C \parallel C$.

With reference to Figure 6(b), it is interesting to observe that the compensating capacitance $C_{\text{comp},j}$ for oscillator $j \in \{1,2\}$ is equal to the product between $C_C \parallel C$ and the difference between the number of connections for oscillator 0, coinciding with the maximum number of connections per oscillator in the unbalanced network of Figure 5(d), and the number of connections for oscillator $j$. Taking inspiration from this finding, for a general unbalanced network with $N$ oscillators, the compensating capacitance $C_{\text{comp},i}$ for oscillator $i \in \{0, 1, \ldots, N-1\}$ is computed via

$$C_{\text{comp},i} \quad = \quad (n_{\max} - n_i) \cdot (C_C \parallel C), \tag{13}$$

where $n_i$ is the number of couplings for oscillator $i$, while

$$n_{\max} \quad = \quad \max_{0 \le i < N} \{n_i\}, \tag{14}$$

is the maximum number of couplings per oscillator in the original network. While reducing the coupling capacitance may allow to accelerate the phase dynamics in the memristive computing engine, a number of factors influence its selection. In this regard, to name but a couple of key aspects, first $C_C$ should not be too small, otherwise the capacitive path between any two oscillators, to be paired so as to reproduce the links, joining the vertices of the associated graph, across the proposed cellular medium, would effectively act as an open circuit. Concurrently, $C_C$ may not be so large as to violate the validity of the approximation $C >> C_C$, used to derive the compensating capacitance $C_{\text{comp},i}$ for each oscillator $i \in \{0, 1, \ldots, N-1\}$ (refer to Equation (13)), which enables to counteract the non-uniformity in the load capacitance per oscillator across the cellular medium, allowing to keep the natural frequencies[11] of the oscillators close together, which facilitates the convergence of the memristive computing engine to some steady state.

In the remainder of this paper, any unbalanced network will first be compensated, and then simulated for the solution of a given graph coloring problem.

### 3.3. Compensation for the Memristor Device-to-Device Variability

Simulating the network of Figure 5(b) for the case, where the device-to-device variability is taken into account, the memristor currents in the two oscillatory circuits may be unable to settle on steady-state oscillatory waveforms. Figure 7(a) shows the current-voltage loci obtained by simulating the memristor model Equations (1), (2), and (3) under a quasi-DC voltage stimulus for each value of the variability parameter $\alpha$ in the set

---

11　The natural frequency of an oscillator is the inverse of the period of the oscillations developing across its circuitry.

$\{0, 0.2, 0.4, 0.6, 0.8, 1.0\}$. It is worth to pinpoint that the distribution of quasi-static characteristics, visualised in this figure, matches the variability observed in analogous loci measured from 196 device samples. Assuming that the oscillator 0 (1) in the two-cell network hosts a memristor, featuring a quasi-DC $i_m$–$v_m$ locus lying in the center (on the right end) of the distribution of Figure 7(a), the network is found to fail to converge to steady-state oscillatory dynamics. This issue is essentially due to the significant mismatch between the DC operating points of the resistive nano-switches in the two oscillators. It may be addressed by reprogramming the DC operating point of the memristor in cell 1. This may be achieved [28] by adjusting either the resistance $R_S$ of the series resistor, as done in this research study, or the voltage $V_S$ of the DC source within the circuit of oscillator 1 until an anti-phase synchronisation pattern is found to emerge in the network.

As demonstrated in the phase diagram visualised in plot (b) of Figure 7, stepping the increment $\Delta R_{S,1}$ in the series resistance $R_S$ of oscillator 1 from $0\,\Omega$ up to $151\,\Omega$, the network keeps featuring non-convergent phase dynamics for a while (see the orange trace relative to the case $\Delta R_{S,1} = 50\,\Omega$). Then, from some point onward, the memristor currents in the two oscillators settle on steady-state oscillatory waveforms, sharing the same frequency, but differing in phase by an offset, which progressively approaches the expected $180°$ level as the series resistance $R_S$ of oscillator 1 gets larger (compare the green and purple traces, obtained for the first and second $\Delta R_{S,1}$-value in the set $\{100, 125\}\,\Omega$, respectively). Finally, the two capacitively-coupled cells attain anti-phase synchronisation (refer to the red trace corresponding to the scenario $\Delta R_{S,1} = 151\,\Omega$).

As another example, Figure 7(c), where the orange and green dashed traces illustrate the time evolution of the phase of oscillators 1 and 2 relative to oscillator 0, respectively, demonstrates that the balanced network of Figure 6(b) fails to converge to a steady-state oscillatory solution when the first, second, and third value in the set $\{0.5, 0, 1\}$ is assigned in turn to the variability parameter $\alpha$ in the model of the memristor in oscillator 0, 1, and 2. A numerical procedure, tuning separately[12], one at a time, the series resistances in oscillators 1 and 2 with the intention to maximise the steady-state phase shifts $\varphi_1^{(s)}$ and $\varphi_2^{(s)}$, determines that decrementing (incrementing) the series resistance of oscillator 1 (2) by $\Delta R_{S,1} = -134\,\Omega$ ($\Delta R_{S,2} = +151\,\Omega$), the network exhibits a steady-state oscillatory pattern characterised by the anti-phase synchronisation between oscillators 1 and 2, on one side, and oscillator 0, on the other side.

In the remainder of this section, in order to compensate for the negative effects that the memristor device-to-device variability has on the performance of a balanced network, the following approach shall be adopted. First, a reference cell, hosting the memristor, to which the random number generator assigns a variability parameter value closest to 0.5 among all $NbO_x$ devices in the array, should be selected. In a hardware implementation of the memristive network, the memristor of the reference oscillator would approximately display the average dynamical behaviour among all the resistive nano-switches employed in the array[13]. This would minimise the subsequent adjustment to be carried out on the bias circuit of each oscillator $j \in \{1, \dots, N-1\}$ to reprogram the DC operating point of the respective memristor[14]. Then, for each $j$-value in the set $\{1, \dots, N-1\}$, the oscillator $j$ is capacitively coupled only to the reference oscillator 0, and the series resistance $R_S$ in its

---

[12] In order to reprogram appropriately the operating point of the memristor in the cell $j \in \{1, 2\}$ of the 3-oscillator network under focus, the cell $j$ itself is capacitively coupled only to the reference cell 0, and, as described earlier, $\Delta R_{S,j}$ is tuned until anti-phase synchronisation emerges in the resulting two-cell network. This procedure is carried out separately for oscillators 1 and 2.

[13] It is important to pinpoint that, while the choice of a reference cell for the preliminary compensation of the memristor device-to-device variability should fall for a specific oscillator, as specified here, no rule dictates the selection of a reference cell for the later computation of the relative phase pattern of the array, as discussed in Section 4.

[14] It is important to observe that, while taking the proposed device-to-device compensation measure, care need to be taken so as to keep the natural oscillation frequency of each oscillator within a close range. In fact, a wide spread in this parameter, inevitably differing across the cellular medium, due to the $R_S$ tuning procedure, would jeopardize the convergence of the bio-inspired computing engine to some steady state.

DC bias circuit is adjusted through a numerical procedure till the point when its increment or decrement by $\Delta R_{s,j}$ induces a 180° phase shift between the memristor currents of the two cells.

It is important to observe that, in hardware, such $R_S$ tuning procedure needs to be carried out only once, during the computing machine testing phase, directly after its fabrication. In order to simplify the programmability of the series resistor, it could be implemented through a voltage-controlled CMOS transistor forced to operate in the linear region.

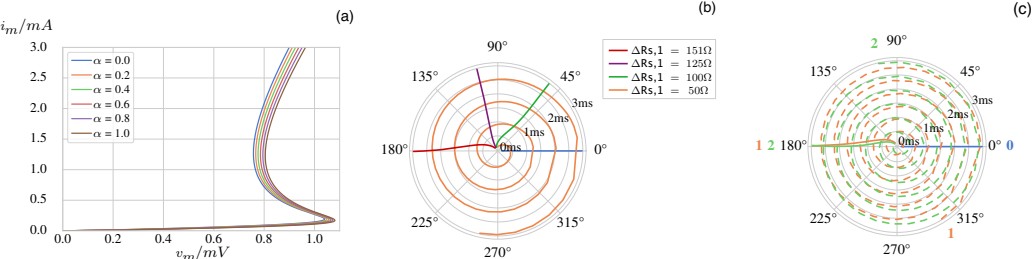

**Figure 7.** (**a**) Spread in the distribution of quasi-DC memristor current-voltage loci, as emerging from numerical simulations of the model Equations (1), (2), and (3) for all values of the variability parameter $\alpha$ in the set $\{0, 0.2, 0.4, 0.6, 0.8, 1.0\}$. (**b**) Phase diagram showing the time evolution of the phase shift of oscillator 1 relative to the 0°-valued phase of the reference oscillator 0 for each of the values of the series resistance increment $\Delta R_{S,1}$ in the set $\{50, 100, 125, 151\}\,\Omega$ (refer in turn to the orange, green, purple, and red traces). The two capacitively-coupled oscillators achieve anti-phase synchronisation for the largest $\Delta R_{S,1}$-value in this set. (**c**) Phase diagram illustrating the phase dynamics of oscillators 1 (in orange) and 2 (in green) for the balanced network of Figure 6(b) for the case where specific parameters in the model of the memristor in the cells 0, 1, and 2 are respectively controlled by the first, second, and third $\alpha$-value within the set $\{0.5, 0, 1\}$ (see the dashed traces), and after the negative effects on the network performance associated to the memristor device-to-device variability have been compensated by incrementing (decrementing) the series resistance $R_S$ by $\Delta R_{S,1} = -134\,\Omega$ ($\Delta R_{S,2} = +151\,\Omega$) (refer to the solid lines).

**Remark 2.** *In view of a future hardware implementation, the parameter setting of the memristive computing engine could be optimized to increase the data processing speed further. However, a comprehensive investigation, aimed to ensure this would not impair the accuracy of the engine calculations, should concurrently be carried out. On one hand, the operating speed of the computing engine should not be so large to prevent the memristor to respond to the stimuli, it experiences over time, which would not allow to exploit thoroughly its rich dynamics for solving the challenging vertex coloring task under focus. On the other hand, the array of capacitively-coupled memristor oscillators is expected to attain a steady state within a sufficiently-short time frame. For all the case studies, investigated in this research work, except for those simulations, in which the proposed computing engine was unable to exit the transient phase, the value assigned to the coupling capacitance $C_C$ enabled the oscillators' phases to converge to steady-state values within tens of milliseconds. Each design parameter may in fact affect some key figure of merit of the bio-inspired network. For example, reducing the capacitance C of each oscillator may induce an acceleration in the phase dynamics. However, it could also reduce the range of admissible memristor NDR bias points, about which the processing unit would undergo limit-cycle oscillations [44], which, as a consequence, would shrink the tuneable range for the series resistance $R_S$ in the device-to-device variability compensation procedure.*

## 4. A Rigorous Strategy for Coloring a Graph via the Network Phase Dynamics

As explained earlier, the assignment of colors to the nodes of a graph is based upon the relative phases among the oscillators of the associated network at steady state. However, a rigorous strategy, as proposed below, needs to be applied at the end of the network

simulation to classify the vertices of the associated graph into color groups, with the intention to use the minimum possible number of colors[15].

As described earlier, the relative phases among the oscillators of a given network are computed at the end of a simulation, on the basis of the time instants at which, within a common steady-state period, the time waveforms of the currents through the memristors cross a threshold value $I_{th}$, here set to 0.5 mA, during the ascending phase. This calculation allows to order the relative phases in increasing order, with the 0° reference level sitting on the first position of the arrangement, which we refer to as *phase shift ordering* in the remainder of the paper. The phase shift ordering directly translates into a corresponding ranking among the oscillators, with those, which feature a lower steady-state phase relative to the reference cell, sitting higher in the table. Equivalently, the ranking among the oscillators may be interpreted as a ranking among the associated vertices.

Our rigorous strategy to assign colors to the vertices of the associated graph on the basis of the network phase dynamics is based upon the analysis of the oscillator/vertex ranking through an iterative procedure composed of $N$ iterations. The first iteration may be summarised as follows. Initially the first vertex in the table, i.e., reference vertex 0, is inserted in the first color group. Proceeding toward the bottom of the table, for $j \in \{2, \ldots, N\}$, vertex at position $j$ in the table is assigned the same color as $(j-1)^{th}$-placed vertex if the graph features no edge between these two vertices, otherwise the $j^{th}$-ranked vertex is defined as the first element of a new color group. After coloring vertex at row $N$ in the table, a final check needs to be carried out to verify if the vertices in the last color group may be merged with those in the first color group. This may be done if and only if no pair of vertices in these two groups is connected by means of an edge in the undirected graph. In cycle $i \in \{2, \ldots, N\}$ of the iterative procedure, the color assignment step is repeated in a similar fashion, analysing progressively the vertices at positions $i, i+1, \ldots, N, 1, 2, \ldots, i-1$ in the table. With such iterative procedure, at least one of the cycles will allow to determine the minimum number of color groups, identifiable by the network, given the phase shift ordering it outputs at the end of a certain simulation. In other words, indicating the $k^{th}$ color group, which, on the basis of the prediction of the $i^{th}$ cycle of the iterative procedure, is identifiable by the network, as $\mathcal{C}_k^{(i)}$ ($k \in \{1, \ldots, m_i\}$, where $m_i \in [n, N]$ denotes the total number of colors assigned to the $N$ nodes of the associated graph in the $i^{th}$ iteration, and $n$ represents the chromatic number of the graph itself, the proposed strategy will output the particular group classification obtained from the $q^{th}$ iteration, whereby $m_q = \min_{i=1}^{i=N} \{m_i\}$.

**Remark 3.** *Remarkably, the initialisation of the oscillatory network, and, particularly, the temporal order of activation of the DC voltage sources in its oscillators, plays a crucial role on the steady-state phase arrangement, and, as a result, on the outcome of the proposed strategy. Consequently, it is possible that the classification of the vertices of a graph into color groups, as estimated via our iterative procedure, is not optimal. Under these circumstances, the network is unable to identify the chromatic number of the associated graph, since it converges to an oscillatory solution corresponding to a local minimum for some optimisation goal associated to the vertex coloring problem.*

Let us denote the *phase shift vector* as $\boldsymbol{\varphi} \triangleq [\varphi_0 = 0°, \ldots, \varphi_{N-1}]$, where $\varphi_i$ stands for the phase shift between oscillator $i$ and reference oscillator 0 over the course of a certain simulation of the network ($i \in \{0, \ldots, N-1\}$). Given that the network of capacitively-coupled oscillators tends to pull the phases of physically-connected cells far apart one

---

[15] The proposed strategy will determine the minimum possible number of color groups, which, under a given initialisation setting, the network is able to identify as it classifies the nodes of the associated graph. Importantly, as will be clarified later, this minimum number does not necessarily coincide with the chromatic number of graph, since the network may converge to a correct but suboptimal solution. Methods allowing the memristive array to overcome a suboptimal solution so as to approach the optimal one will be presented shortly.

from the other, it is expected that the phase dynamics unfold toward a steady-state pattern maximising a function $F(\boldsymbol{\varphi})$ of the form

$$F(\boldsymbol{\varphi}) \triangleq \frac{1}{2} \cdot \sum_{i=0}^{N-1} \sum_{j=0}^{N-1} a_{i,j} \cdot |\varphi_i - \varphi_j|, \tag{15}$$

where $a_{i,j}$ is the element at row $i$ and column $j$ of the so-called *adjacency matrix* **A**, which encodes the coupling arrangement within a $N$-node graph[16]. Transforming the expression for $F(\boldsymbol{\varphi})$ in Equation (15), another function of the phase shift vector, defined as

$$G(\boldsymbol{\varphi}) \triangleq \frac{1}{2} \cdot \sum_{i=0}^{N-1} \sum_{j=0}^{N-1} a_{i,j} \cdot \cos(\varphi_i - \varphi_j), \tag{16}$$

is chosen to describe the *optimisation goal* of the network as it evolves toward a stable solution. It is worth pinpointing that the phase shift vector $\boldsymbol{\varphi}^{(s)} = [\varphi_0^{(s)} = 0°, \ldots, \varphi_{N-1}^{(s)}]$, emerging in a well-behaved network at steady state, minimises the function $G(\boldsymbol{\varphi})$. For a general network, however, the optimization goal function might feature a number of local minima besides the global minimum. If the oscillatory solution, the network converges to, at the end of a certain simulation, corresponds to a local (the global) minimum of $G(\boldsymbol{\varphi})$, the application of the vertex coloring strategy, described earlier, results in a group number higher than (equal to) the chromatic number of the associated graph.

The analysis of an exemplary network, specifically the one shown in Figure 3(b), allows to gain a deeper insight into the phase dynamics of a memristive array in a pair of simulation scenarios, associated to a suboptimal and to the optimal initialisation scenario, respectively. The network under focus is already balanced, since each of its oscillators is coupled to the 2 adjacent cells, thus only the memristor device-to-device variability needs to be neutralised. Figure 8(a) ((c)) visualises the time evolution of the phase dynamics emerging in the network under a suboptimal (the optimal) initialisation setting. In both scenarios the optimisation goal function decreases over time, but, in the first (latter) case, $G(\boldsymbol{\varphi})$ converges asymptotically toward a local (the global) minimum, as depicted in plot (b) ((d)) of the same figure.

---

[16]   If an (no) edge connects vertices $i$ and $j$, then $a_{i,j} = a_{j,i} = 1(0)$. Note that $\mathbf{A} = \mathbf{A}^T \in \mathbb{R}^{N \times N}$.

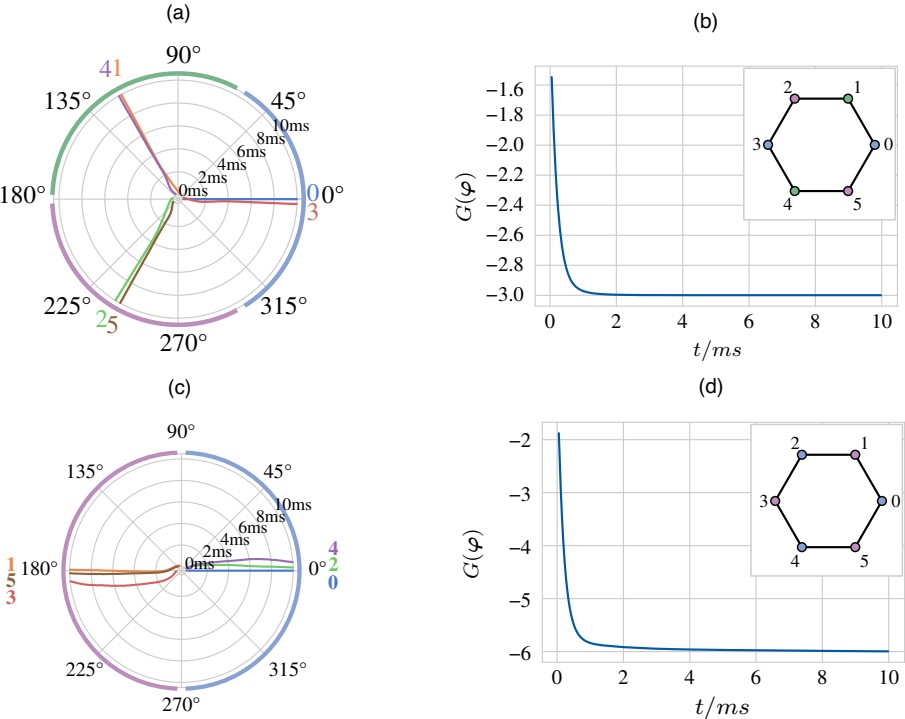

**Figure 8.** (**a**) ((**c**)) Phase dynamics of the balanced network of Figure 3(b) with compensation for the memristor device-to-device variability and under a suboptimal (the optimal) initialisation setting. (**b**) ((**d**)) Time evolution of the optimisation goal function toward a local (the global) minimum in the simulation scenario illustrated in plot (**a**) ((**c**)). In the first (latter) case the application of the vertex coloring strategy to the respective vertex ranking divides the 6 nodes of the graph of Figure 3(a) into 3 (2) colors. In the first (latter) case the composition of each of the 3 (2) color groups is made clear by the colors assigned in plot (**a**) ((**c**)) to the arcs of the circular sectors, which host the final destinations of the traces associated to phase shifts clustering together, as well as by the colors assigned in the inset of plot (**b**) ((**d**)) to the respective nodes of the graph. Since the chromatic number of the graph is 2, the relative phases among the oscillators of the network converge to a suboptimal (the optimal) pattern in the simulation of plot (**a**) ((**c**)).

Let us gain a deeper insight into the outcome of the proposed graph coloring strategy for the case of the suboptimal simulation, which, as demonstrated in Figure 8(a), outputs the following steady-state phase shift vector:

$$\boldsymbol{\varphi}^{(s)} = [\varphi_0^{(s)}, \varphi_1^{(s)}, \varphi_2^{(s)}, \varphi_3^{(s)}, \varphi_4^{(s)}, \varphi_5^{(s)}]^{\mathrm{T}} = [0°, 118°, 240°, 358°, 120°, 242°]^{\mathrm{T}}. \quad (17)$$

The resulting phase shift ordering, namely $\varphi_0^s - \varphi_1^s - \varphi_4^s - \varphi_2^s - \varphi_5^s - \varphi_3^s$, allows to establish the following vertex ranking[17]:

$$0 - 1 - 4 - 2 - 5 - 3. \quad (18)$$

---

[17] Since, here, oscillator $i$ is associated to vertex $i$ for each $i \in \{0, \dots, N-1\}$, the oscillator sequence, corresponding to the phase shift ordering, may be indifferently referred to as oscillator ranking or vertex ranking. As will be clarified later on, this is not always the case, when perturbation actions are performed on the network to enhance its performance.

For each $i$-value in the set $\{1, \ldots, 6\}$, the $i^{\text{th}}$ cycle of the proposed iterative graph coloring procedure provides the following group classification of the nodes of the undirected graph of Figure 3(a):

$$\mathcal{C}_1^{(1)} = \{0, 3\}, \mathcal{C}_2^{(1)} = \{1, 4\}, \mathcal{C}_3^{(1)} = \{2, 5\}, \tag{19}$$

$$\mathcal{C}_1^{(2)} = \{1, 4\}, \mathcal{C}_2^{(2)} = \{2, 5\}, \mathcal{C}_3^{(2)} = \{3, 0\}, \tag{20}$$

$$\mathcal{C}_1^{(3)} = \{4, 2\}, \mathcal{C}_2^{(3)} = \{5, 3\}, \mathcal{C}_3^{(3)} = \{0\}, \mathcal{C}_4^{(3)} = \{1\}, \tag{21}$$

$$\mathcal{C}_1^{(4)} = \{2, 5\}, \mathcal{C}_2^{(4)} = \{3, 0\}, \mathcal{C}_3^{(4)} = \{1, 4\}, \tag{22}$$

$$\mathcal{C}_1^{(5)} = \{5, 3\}, \mathcal{C}_2^{(5)} = \{0\}, \mathcal{C}_3^{(5)} = \{1, 4\}, \mathcal{C}_4^{(5)} = \{2\}, \tag{23}$$

$$\mathcal{C}_1^{(6)} = \{3, 0\}, \mathcal{C}_2^{(6)} = \{1, 4\}, \mathcal{C}_3^{(6)} = \{2, 5\}. \tag{24}$$

In each of cycles 1, 2, 4, and 6, our iterative procedure assigns the vertices of the graph of Figure 3(a) a minimum number of colors, i.e., 3, which, as expected, is higher than the chromatic number of the graph itself, i.e., $n = 2$. In each of these iterations the same three pairs of vertices are grouped together, thus any number in the set $\{1, 2, 4, 6\}$ may be assigned to $q$, and, as a result, any of Equations (19), (20), (22), and (24) may be taken as outcome of the graph coloring procedure. The suboptimal solution of the graph coloring problem is clearly indicated in Figure 8(a), where 3 different colors are used to mark the arcs of the 3 circular sectors, which in turn host the final destinations of the 2 traces associated to the phase shifts of the oscillator pairs $(0, 3)$, $(1, 4)$, and $(2, 5)$, as well as in the inset of Figure 8(b), where a distinct color is used to fill each node pair in the set $\{(0, 3), (1, 4), (2, 5)\}$.

Let us now analyse comprehensively the working principles of our graph coloring strategy for the case of the optimal simulation, which produces Figure 9(a) for the dynamical behaviour of the memristor currents in the 6 oscillators of the network over the steady-state time interval $t \in [9.9, 10]$ ms. Looking in more detail at the evolution of the memristor currents over the last part of this time interval, i.e., for $t \in [9.97, 10]$ ms, Figure 9(b) shows the common period[18] $T$ of the 6 oscillatory waveforms, and marks the time instants $t_0$ and $t_3$, at which $i_{m,0}$ and $i_{m,3}$ cross the threshold value $I_{th} = 0.5$ ms, respectively, allowing to compute the steady-state phase shift $\varphi_3(s)$ between cells 3 and 0, as described in the figure caption. Computing also the relative phase shifts of the other 5 oscillators of the network over the common $T$-long time interval shown in plot (b) of Figure 9(b), the steady-state phase shift vector $\boldsymbol{\varphi}^{(s)}$ is found to be equal to

$$\boldsymbol{\varphi}^{(s)} = [\varphi_0^{(s)}, \varphi_1^{(s)}, \varphi_2^{(s)}, \varphi_3^{(s)}, \varphi_4^{(s)}, \varphi_5^{(s)}]^{\text{T}} = [0°, 180°, 5°, 195°, 11°, 182°]^{\text{T}}. \tag{25}$$

as indicated in Figure 8(c). The resulting phase shift ordering, namely $\varphi_0^{(s)} - \varphi_2^{(s)} - \varphi_4^{(s)} - \varphi_1^{(s)} - \varphi_5^{(s)} - \varphi_3^{(s)}$, allows to establish the following vertex ranking:

$$0 - 2 - 4 - 1 - 5 - 3. \tag{26}$$

For each $i$-value in the set $\{1, \ldots, 6\}$, the $i^{\text{th}}$ cycle of the proposed iterative graph coloring procedure provides the following group classification of the nodes of the undirected graph of Figure 3(a):

---

[18] in this work the estimation of the common period $T$ of the oscillations developing in a $N$-cell network, and the associated group ordering of the phase shifts of the cells 1, ..., $N - 1$ relative to the null phase of the reference cell 0 are carried out every cycle throughout the duration of any simulation.

$$
\begin{aligned}
C_1^{(1)} &= \{0, 2, 4\}, \; C_2^{(1)} = \{1, 5, 3\}, & (27) \\
C_1^{(2)} &= \{2, 4, 0\}, \; C_2^{(2)} = \{1, 5, 3\}, & (28) \\
C_1^{(3)} &= \{4, 1\}, \; C_2^{(3)} = \{5, 3\}, \; C_3^{(3)} = \{0, 2\}, & (29) \\
C_1^{(4)} &= \{1, 5, 3\}, \; C_2^{(4)} = \{0, 2, 4\}, & (30) \\
C_1^{(5)} &= \{5, 3, 1\}, \; C_2^{(5)} = \{0, 2, 4\}, & (31) \\
C_1^{(6)} &= \{3, 0\}, \; C_2^{(6)} = \{2, 4\}, \; C_3^{(6)} = \{1, 5\}. & (32)
\end{aligned}
$$

In each of cycles 1, 2, 4, and 5, our iterative procedure assigns the vertices of the graph of Figure 3(a) a minimum number of colors, i.e., 2, which, as expected, coincides with the chromatic number of the graph itself, i.e., $n = 2$. In each of these iterations the same two triplets of vertices are grouped together, thus any number in the set $\{1, 2, 4, 5\}$ may be assigned to $q$, and, as a result, any of Equations (27), (28), (30) and (31) may be taken as outcome of the graph coloring procedure. The optimal solution of the graph coloring problem is clearly indicated in Figure 8(c), where 2 different colors are used to mark the arcs of the 2 circular sectors, which in turn host the final destinations of the 3 traces associated to the phase shifts of the oscillator triplets $(0, 2, 4)$, and $(1, 5, 3)$, as well as in the inset of Figure 8(d), where a distinct color is used to fill each node triplet in the set $\{(0, 2, 4), (1, 5, 3)\}$.

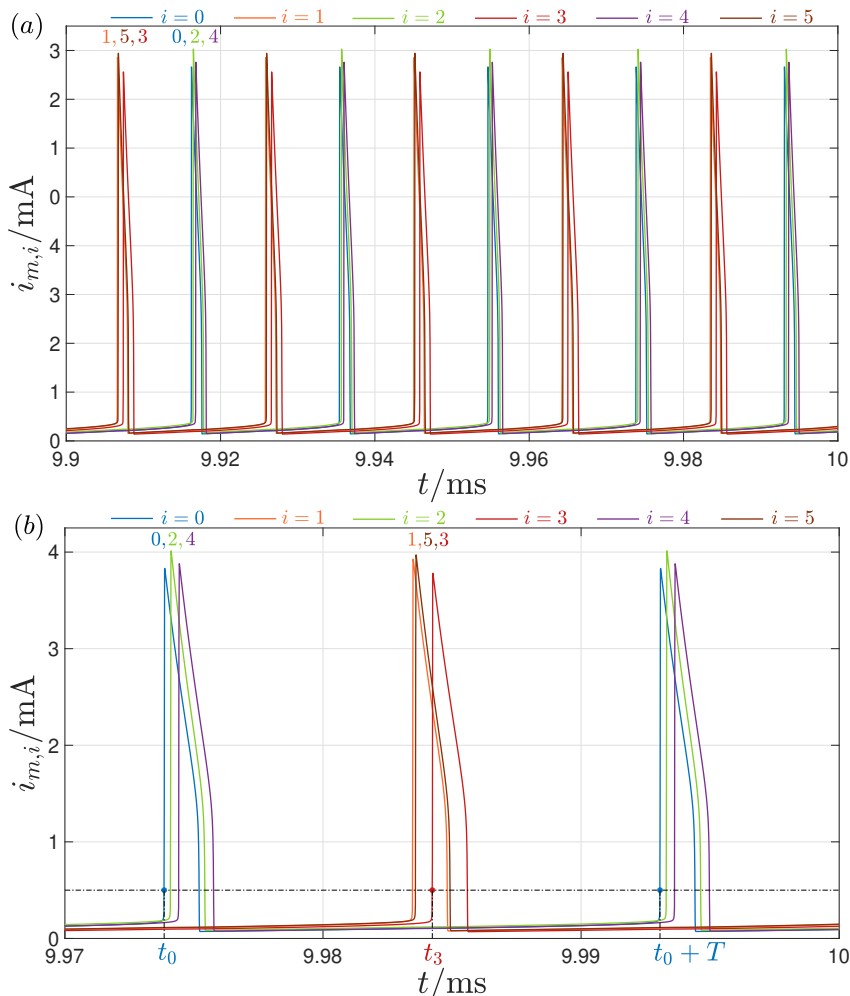

**Figure 9.** (**a**) Steady-state time evolution of the memristor current $i_{m,i}$ of oscillator $i \in \{0, 1, 2, 3, 4, 5\}$ over the time interval $[9.9, 10]$ ms for the simulation of the balanced network of Figure 3(b) with compensation for the memristor device-to-device variability and under the optimal initialisation setting (see also Figure 8(c),(d) for more results obtained from this simulation). The differences in the peak values of the waveforms originate from the variability in the static and dynamic properties of the samples, as reproduced by our memristor model. As was already shown in relation to their relative phases in Figure 8(c), the oscillator triplets $(1, 5, 3)$ and $(0, 2, 4)$ group together, as respectively indicated over the first and second half of the first observable cycle, where the order of appearance of the nearby peaks of the memristor currents follows in turn the patterns 1-5-3 and 0-2-4. The same color coding map, as established in Figure 8(c) and reused in the inset of Figure 8(d), is adopted here to differentiate between the traces pertaining to distinct oscillators. (**b**) Zoom-in view of the time behaviour of each memristor current in the network across the time span $[9.97, 10]$ ms. The common period $T$ of the oscillatory waveforms is found to be equal to $19.21\,\mu s$. As an example, the time instant $t_0$ $(t_3)$, at which the memristor current $i_{m,0}$ $(i_{m,5})$ of oscillator 0 (3) crosses the threshold $I_{th} = 0.5\,\text{mA}$ in its ascending phase over the first observable cycle, gives $9973.8457\,\mu s$ $(9984.2351\,\mu s)$, allowing to compute the steady-state phase of oscillator 3 relative to the reference oscillator 0 via $\varphi_3^{(s)} = \Delta t_3 \cdot \omega_0$, with $\omega_0 = \frac{2 \cdot \pi}{T}$. Performing a calculation of this kind for each of the remaining 5 oscillators results in the steady-state phase shift vector $\boldsymbol{\varphi}^{(s)}$ reported in Equation (25).

## 5. Control Paradigms to Resolve Local Minima-Based Impasse Conditions

In order to overcome the impasse, emerging when a memristive network converges to a stable oscillatory solution associated to a local minimum of the optimisation goal function $G(\boldsymbol{\varphi})$, it is necessary to destabilise the array through an ad-hoc perturbation. We identified two possible strategies allowing to pull the dynamical system out of a local minimum of

the respective optimisation problem. In many cases, after recovering from the impasse, the network was found to converge asymptotically toward an oscillatory solution associated the global minimum of the optimisation goal function. The proposed approaches, referred to as *crossover* and *pulse destabilisation* strategies, are presented in the following two sections.

### 5.1. Crossover Strategy

One of the strategies, allowing the network to overcome an impasse situation, inspired from genetic algorithms [45], is based upon the interchange between the connections of two properly-selected oscillators[19] ([31,33]). This corresponds to the exchange of the associations between the two cells of the network and the corresponding vertices in the relative graph. In order to select the most appropriate pair of oscillators—let us use indices $i$ and $j$ to label them—for the crossover, the following two-step procedure is applied.

1. For each value of $k$ in the set $\{0, \ldots, N-1\}$, the vertex $k$ is removed from the original $N$-node graph, and the iterative vertex coloring strategy is applied to the resulting graph of $(N-1)$ nodes, using a modified version of the vertex ranking, which is tabulated beforehand, after a simulation of the oscillatory network, under a generic sub-optimal initialisation setting, attains the steady state. Specifically, the label of the vertex $k$, taken out of the original graph, is removed from the original vertex ranking, resulting in a new table with $N-1$ entries. For each value of $k$, a $N \times N$ matrix, denoted as $\mathbf{A}^{(k)}$, and obtained from the original adjacency matrix $\mathbf{A}$ by setting to 0 all the elements at row $k$ and at column $k$, may still be used to define the connectivity of the respective $(N-1)$-node graph. Coloring the $N-1$ vertices of $N$ distinct graphs, at least one of the $N$ problems will be found to admit the best solution, allowing to categorise the $N-1$ nodes of the relative graph through the lowest number of color groups. The particular node $k$, which, extracted out of the original graph, allows the resulting network to identify the least number of colors according to our iterative vertex coloring procedure, may then be chosen as first vertex $i$ to involve in the crossover[20].

2. Assigning, one at a time, any integer from the set $\{0, \ldots, i-1, i+1, \ldots, N-1\}$ to $k$, the iterative vertex coloring strategy is then applied to a new vertex ranking, obtained from the original table by interchanging the positions of vertices $i$ and $k$. Note that the original $N$-node graph, with connectivity defined by the adjacency matrix $\mathbf{A}$, should be considered in each of the $N-1$ applications of the iterative vertex coloring strategy, since nothing else, except for the correspondence between oscillators and vertices, is affected in a crossover operation[21]. Solving the resulting $N-1$ vertex coloring problems, the solution, assigning the least number of colors to the $N$ nodes of the original graph, will be determined. It may happen that, on the basis of the proposed iterative procedure, for two or more values of $k$, the exchange between the positions of oscillators $k$ and $i$ in the original vertex ranking results in a common lowest number of color groups for the $N$ nodes of the original graph. In this case, the choice of the second oscillator $j$ to involve in the crossover falls for the particular candidate

---

[19] The application of a crossover to pairs of oscillators implies the necessity to endow the network with reprogrammable connections, e.g. via transistor-based switches, which, however, would add on to the integrated circuit (IC) overhead in a future hardware implementation of the network.

[20] In fact, it is highly probable that this node mostly prevents the optimisation measure of Equation (16) from attaining the global minimum, which would provide as solution to the vertex coloring task the chromatic number of the original $N$-node graph, as desired. In case, for each of two or more values of $k$, the application of our vertex coloring strategy to the respective $(N-1)$-node graph, obtained by removing the vertex $k$ from the original graph, results in a common lowest number of colors, any of these node $i$ candidates may be finally considered for the crossover

[21] The interchange between nodes $i$ and $k$ operated on the original vertex ranking is due to the fact that the application of a crossover between the corresponding oscillators in the network is equivalent to exchanging their associations to the respective pair of vertices in the original graph. The relative phases, inherent to the oscillators, maintain the same ordering, as established originally. As a result, the oscillator ranking remains unaltered, but the mapping from oscillator ranking to vertex ranking is subject to the earlier mentioned node interchange.

$k$, whose relative phase $\varphi_k$ features the largest distance from the relative phase $\varphi_i$ of oscillator $i$ in the steady-state phase shift vector $\boldsymbol{\varphi}$ obtained through the simulation preceding the application of the two-step strategy.

In order to gain a deeper understanding of the proposed two-step procedure, let us apply it to the vertex ranking $0 - 1 - 4 - 2 - 5 - 3$, obtained at steady state from a simulation of the balanced network of Figure 3(b) (or, equivalently, of Figure 10(b), the corresponding graph of which is illustrated again in Figure 10(a) for the sake of clarity) with compensation for the memristor device-to-device variability and under the same sub-optimal initialisation setting as in the simulation illustrated in Figure 8(a),(b) (see the caption of Figure 11 for details). According to the first step of the procedure, applying our iterative vertex coloring strategy to the original vertex ranking, i.e., $0 - 1 - 4 - 2 - 5 - 3$, after depriving it of the $k^{th}$ node label, with the associated 5-node graph obtained from the original one of Figure 3(a) by breaking all the connections of the $k^{th}$ vertex ($k \in \{0, 1, 2, 3, 4, 5\}$), the smallest number of colors, the vertices of the 5-node graph are assigned to, is 2 for each $k$-value in the set $\{0, 1\}$, and 3 for each $k$-value in the set $\{2, 3, 4, 5\}$. The choice for the $i$ cell for the crossover may then fall either on reference oscillator 0 or on oscillator 1. Let us choose the latter cell. In line with the second step of the procedure, exchanging the positions of labels $k$ and $i = 1$ in the original vertex ranking $0 - 1 - 4 - 2 - 5 - 3$ ($k \in \{0, 2, 3, 4, 5\}$), the application of the iterative vertex coloring strategy to the resulting sequence, with respect to the original 6-node graph of Figure 3(a), results in a minimum number of color groups equal to 2 for each $k$-value in the set $\{0, 2\}$, to 3 for each $k$-value in the set $\{3, 4\}$, and to 4 for $k = 5$. Now, given that, in the original steady-state phase shift ordering, $\varphi_2^{(s)} - \varphi_0^{(s)} = 122° > \varphi_1^{(s)} - \varphi_0^{(s)} = 118°$, oscillator with label $k = 2$ is selected as cell $j$ for the crossover. With reference to Figure 10, where plots (a) and (b) show once again the six-node ring-based graph, and the associated capacitively-coupled array of memristor oscillators, plot (c), redrawn in a different but equivalent form in plot (d), depicts the novel coupling arrangement in the network upon the interchange between the connections of oscillators 1 and 2.

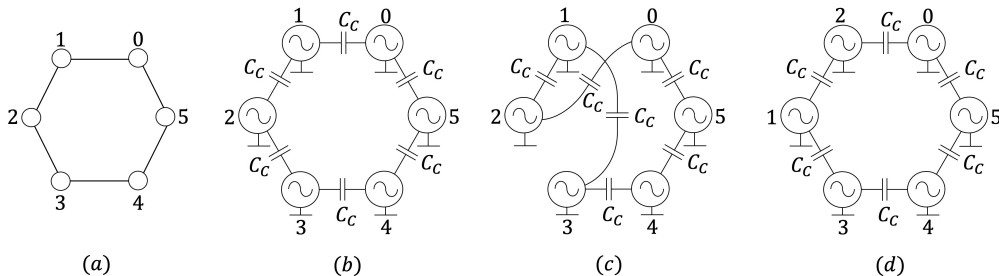

(a)     (b)     (c)     (d)

**Figure 10.** (**a**) Original 6-node ring-based graph. (**b**) ((**c**) or, equivalently, (**d**)) Coupling arrangement in the network associated to the graph in (**a**), before (after) a crossover between cells 1 and 2, which swaps the correspondence between these cells and the respective nodes in the graph.

The numerical results illustrated in Figure 11, where plots (a), (b), and (c) respectively show phase dynamics, time evolution of the optimisation goal function, and temporal trend of the solution of the classification task, respectively, provide evidence for the success of the circuit implementation of the crossover strategy in pulling the dynamical system out of the impasse state, allowing its asymptotic convergence to the solution of the graph coloring problem associated to the global minimum of $G(\boldsymbol{\varphi})$. These results were obtained by simulating the balanced network of Figure 10(b), with compensation for the memristor device-to-device variability, and under the same sub-optimal initialisation setting as in the simulation illustrated in Figure 8(a),(b). With reference to Figure 11, at the end of the first part of the simulation, covering the time span $t \in [0, 5)$ ms, the optimisation goal function was found to sit at a local minimum value, specifically $-3$ (see plot (b)), and the minimum number of colors, which, on the basis of our iterative vertex coloring procedure, may be

assigned to the nodes of the graph in Figure 10(a), is 3 (refer to plot (c)). At $t = 5$ ms the connections of oscillators 1 and 2 were interchanged, as shown in plot (c) or, equivalently, in plot (d) of Figure 10. Looking more at Figure 11, the dynamics of the relative phases of the cells resume directly after the crossover, approaching a new stable steady-state pattern, whereby $G(\boldsymbol{\varphi})$ is found to sit at its global minimum level, particularly $-6$ (see plot (b))), and the network is able to identify the chromatic number of the graph in Figure 10(a), as determined through the proposed iterative vertex coloring procedure (refer to plot (c)).

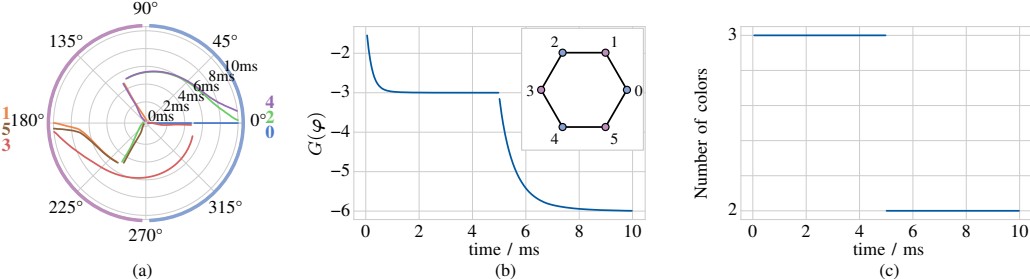

(a)                          (b)                          (c)

**Figure 11.** (**a**) Time evolution of the relative phases of the oscillators of the balanced network of Figure 3(b), with compensation for the memristor device-to-device variability, and under the same sub-optimal initialisation setting as in the simulation illustrated in Figure 8(a),(b), for the case where the connections of oscillators 1 and 2 are interchanged at $t = 5$ ms. Right before the application of the crossover to the two cells, the network is found to sit on a stable oscillatory solution associated to a local minimum of the optimisation goal function (see also the three phase clusters emerging in plot (**a**) right before the crossover procedure). Despite the phase shift vector, measured much earlier than it was done in the simulation of Figure 8(a),(b), was found to be slightly different from the one reported in Equation (17), specifically $\boldsymbol{\varphi}^{(s)} = [0°, 118°, 238°, 359°, 119°, 240°]^{\mathrm{T}}$, the resulting vertex ordering remains defined by Equation (18). (**b**) Evolution of the optimisation goal function over time. (**c**) Minimum number of color groups assigned through the iterative vertex coloring procedure of Section 4 to the nodes of the graph in Figure 3(a) over time (the procedure is applied once every $T$-long cycle, with the common period $T$ of the oscillatory waveforms of the currents through the memristors, measured over the time interval $[4.965, 4.984]$ ms, found to be equal to 19.24 μs). After the crossover these nodes are classified into 2 groups (**c**). The interchange between the couplings of oscillators 1 and 2 was thus found to resolve the impasse, allowing the memristive array to approach the optimal solution associated to the global minimum of $G(\boldsymbol{\varphi})$, and to identify the chromatic number $n = 2$ of the associated graph (see also the two phase clusters emerging in plot (**a**) at the end of the simulation).

As anticipated earlier, the circuit implementation of the crossover strategy crucially requires the availability of reprogrammable connections among the oscillators of the network. This inevitably increases the area overhead of the hardware platform. Furthermore, the circuitry necessary to endow the network with reprogrammable connectivity, may introduce some mismatch between the capacitive loads of the oscillators, which may represent an obstacle toward the convergence of the phase dynamics of the memristive array toward the pattern corresponding to the global minimum of the optimisation problem. For this reason, another strategy for pulling the dynamical system out of a local minimum, which is more amenable to circuit implementation, is presented in the next section.

### 5.2. Pulse Destabilisation Strategy

Inspired from *simulated annealing algorithms* [46] as well from our latest approach—referred to as *Kick-Fly-Catch* paradigm—for humanoid robot motion control [47,48], the proposed strategy ([31,33]) is based upon supplying energy to the system, while it sits in an impasse state. Particularly, a pulse is applied to an ad-hoc oscillator by offsetting the value $V_S$ of its DC voltage source by an appropriate amount $\Delta V_S$ for a given time interval of length $T_p$. Consequently, the relative phase of the oscillator, sitting at the steady-state

level $\varphi^{(s)}$ previous to the stimulation, experiences a sudden shift by $\Delta\varphi$, which is found to depend upon amplitude $\Delta V_S$ and length $T_p$ of the voltage pulse. Taking inspiration from the research study presented in [49], extensive numerical simulations revealed that, for a given pulse width[22] $T_p$, there is an approximately-linear relation between the pulse height $\Delta V_S$ and the sudden shift $\Delta\varphi$, that the relative phase of the oscillator experiences upon destabilisation, i.e.,

$$\Delta V_S \approx \frac{V_0 \cdot \Delta\varphi}{180°}, \tag{33}$$

where $V_0$ was numerically set to $-0.23\,\text{V}$ for the networks analysed in this research study[23]. Provided the right choice is made regarding the cell to perturb, and the proper amplitude and width are assigned to the voltage pulse stimulus, the resulting resumption of the phase dynamics of the oscillators should ideally allow the optimisation goal function to move out of the local minimum, converging asymptotically toward its global minimum. A two-step procedure, presented below, is proposed here to determine the most suitable cell to perturb and the most appropriate pulse amplitude.

1. The most suitable oscillator $i \in \{0, \ldots, N-1\}$ to target in the pulse destabilisation action is determined in the same way as was done for the selection of the cell $i \in \{0, \ldots, N-1\}$ to involve in the crossover process (see the first step in the procedure aimed to choose the right cell pair $(i, j)$ to involve in the coupling interchange strategy).

2. The second step is aimed to determine the appropriate shift $\Delta\varphi$ to be added to the steady-state relative phase $\varphi_i^{(s)}$ of oscillator $i$ for pulling the network out of the local minimum state, facilitating its convergence to an oscillatory solution, which would ideally correspond to the least number of color groups for the $N$ vertices of the associated graph. To accomplish this task, for each value of $k$ within the set $\{1, \ldots, M-1\}$, with M a predefined positive integer, the offset $\Delta\varphi_k = k \cdot \frac{360°}{M}$ is added to the phase shift $\varphi_i^{(s)}$ of cell $i$ in the steady-state relative phase shift vector $\boldsymbol{\varphi}^{(s)}$ recorded before the application of the pulse destabilisation process, and the iterative graph coloring procedure is applied to the resulting vertex ranking for the original graph. The choice for the most appropriate offset $\Delta\varphi$, within the specified set of $k$-dependent uniformly-spaced values, goes for the $\Delta\varphi_k$-candidate, which, according to our graph coloring strategy, allows the network to classify the nodes of the associated graph in the lowest number of color groups. If, for two or more $k$-values, the application of the iterative graph coloring procedure to the vertex ranking, resulting from the phase shift ordering, obtained by adding up the relevant offset $\Delta\varphi_k$ to the phase shift $\varphi_i^{(s)}$ of oscillator $i$ in the steady-state relative phase shift vector $\boldsymbol{\varphi}^{(s)}$, leads to the identification of the same lowest number of colors, the selection goes for the $\Delta\varphi_k$-candidate featuring the largest modulus. Finally, the pulse amplitude of the $T_p$-long stimulus to be applied to oscillator $i$ is obtained from Equation (33).

In order to gain more insights into the mechanisms underlying this two-step procedure, let us apply it to the vertex ranking $0 - 1 - 4 - 2 - 5 - 3$, obtained at steady state from a simulation of the balanced network of Figure 3(b) with compensation for the memristor device-to-device variability and under the same sub-optimal initialisation setting as in the simulation illustrated in Figure 8(a),(b) (see the caption of Figure 12 for details). Given that the first step of the procedure is identical to the first step of the algorithm allowing to

---

[22] In this work $T_p$ was set to twice the common graph-dependent period $T$ of the oscillatory waveforms of the capacitor voltages and of the memristor currents in the network before the application of the pulse destabilisation paradigm.

[23] We acknowledge, however, that the most suitable formula, expressing the relationship between the amplitude $\Delta V_S$ of a destabilising pulse of fixed width $T_p$ and the resulting sudden shift $\Delta\varphi$ in the phase of the perturbed oscillator, may depend upon network properties and parameters. A deeper study, aimed to optimise the shape of the destabilisation stimulus, will be carried out in the future.

select the optimal cell pair $(i, j)$ to involve in the crossover strategy, retrieving the results presented in the previous section, either oscillator from the label set $i \in \{0, 1\}$ may be chosen as target of the pulse destabilisation action. Let us go for the latter one. Following the guidelines established for the second step of the procedure, setting $M$ to 4, for each $k$-value in the set $\{1, 2, 3\}$, colors are assigned to the nodes of the original graph of Figure 3(a) by applying the proposed iterative methodology to the $k^{\text{th}}$ vertex ranking variant derived from the phase shift vector $\boldsymbol{\varphi}^{(s)} = [0°, 118°, 238°, 359°, 119°, 240°]^{\text{T}}$ recorded right before the time instant[24] $t = 5\,\text{ms}$, at which the pulse perturbation action is commenced by adding up the $k^{\text{th}}$ value of $\Delta \varphi_k$ in the set $\{90°, 180°, 270°\}$ to the relative phase $\varphi_1^{(s)}$ of oscillator 1. Analysing the $k^{\text{th}}$ vertex ranking within the set $\{0 - 4 - 1 - 2 - 5 - 3, 0 - 4 - 2 - 5 - 1 - 3, 0 - 1 - 4 - 2 - 5 - 3\}$ $(k \in \{1, 2, 3\})$, according to our node coloring paradigm the network identifies a minimum number of color groups equal to the $k^{\text{th}}$ number in the set $\{3, 2, 3\}$. Setting $k$ to 2 in the formula for $\Delta \varphi_k$, from Equation (33) the amplitude $\Delta V_S$ of the pulse applied to oscillator 1 at $t = 5\,\text{ms}$ for a $T_p = 38.48\,\mu\text{s}$-long time interval is set to $-0.23\,\text{V}$ (see the caption of Figure 12 for details).

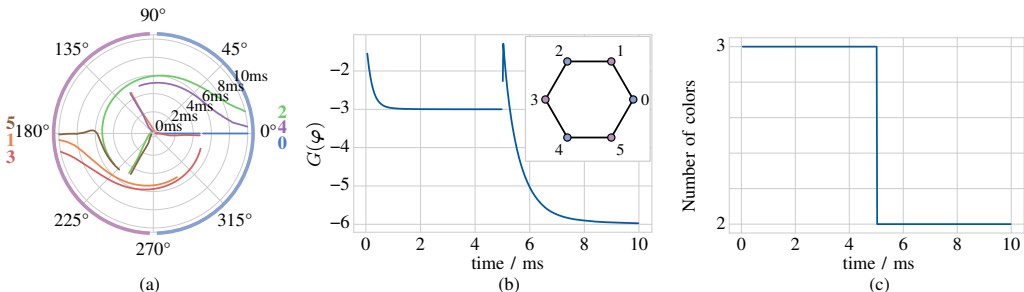

(a)　　　　　　(b)　　　　　　(c)

**Figure 12.** (**a**) Phase dynamics of the balanced network of Figure 3(b), with compensation for the memristor device-to-device variability, and under the same sub-optimal initialisation setting as in the simulation illustrated in Figure 8(a)-(b), for the case where the voltage $V_S$ of the DC source in oscillator 1 is offset by $\Delta V_s = -0.23\,\text{V}$ from the time instant $t = 5\,\text{ms}$ for a temporal window of duration $T_p = 38.48\,\mu\text{s}$ (the common period $T$ of the oscillatory waveforms of the currents through the memristors, measured over the time interval $[4.965, 4.984]\,\text{ms}$, was found to be equal to $19.24\,\mu\text{s}$). As discussed earlier, right before the application of the pulse to cell 1, which triggers a sudden shift in its relative phase by approximately $180°$, the network is found to sit on a stable oscillatory solution associated to a local minimum of the optimisation goal function (see also the three phase clusters emerging in plot (**a**) right before the pulse destabilisation action). Despite the phase shift vector, measured much earlier than it was done in the simulation of Figure 8(a),(b), was found to be slightly different from the one reported in Equation (17), specifically $\boldsymbol{\varphi}^{(s)} = [0°, 118°, 238°, 359°, 119°, 240°]^{\text{T}}$, the resulting vertex ordering remains defined by Equation (18). (**b**) Time evolution of the optimisation goal function. (**c**) Minimum number of color groups assigned through the iterative vertex coloring procedure of Section 4 to the nodes of the graph in Figure 3(a) over time (the procedure is applied once every $T$-long cycle). After the pulse destabilisation 2 colors are assigned to these nodes. The pulse-based perturbation of oscillator 1 was thus found to resolve the impasse, allowing the memristive array to approach the optimal solution associated to the global minimum of $G(\boldsymbol{\varphi})$, and to identify the chromatic number $n = 2$ of the associated graph (see also the two phase clusters emerging in plot (**a**) at the end of the simulation). Despite, at the time instant $t = 5\,\text{ms}$, when the pulse perturbation commences, $G(\boldsymbol{\varphi})$ undergoes a sudden increase from the local minimum value of $-3$, it descends steeply straight away, decreasing monotonically toward the global minimum value of $-6$ thereafter.

The numerical results illustrated in Figure 12, where plots (a), (b), and (c) respectively show phase dynamics, time evolution of the optimisation goal function, and temporal trend

---

24　In order to present a fair comparison between the beneficial effects of the crossover and pulse destabilisation control paradigms, we ensured that the simulations in Figures 11 and 12 provided identical results for $t \in [0, 5)\,\text{ms}$ by choosing the same initialisation setting, and assigning a common random set of $\alpha$-values to the memristors.

of the solution of the classification task, respectively, provide evidence for the success of the circuit implementation of the pulse destabilisation strategy in pulling the dynamical system out of the impasse state, allowing its asymptotic convergence to the solution of the graph coloring problem associated to the global minimum of $G(\boldsymbol{\varphi})$. These results were obtained by simulating the balanced network of Figure 10(b), with compensation for the memristor device-to-device variability, and under the same sub-optimal initialisation setting as in the simulation illustrated in Figure 8(a),(b). With reference to Figure 12, at the end of this first part of the simulation, covering the time interval expressed as $t \in [0, 5)$ ms, the optimisation goal function was found to sit at a local minimum value, specifically $-3$ (see plot (b)), and the minimum number of colors, which, on the basis of our iterative vertex coloring procedure, may be assigned to the nodes of the graph in Figure 3(a), is 3 (refer to plot (c)). From the time instant $t = 5$ ms and for a $T_p = 38.48\,\mu\text{s}$-long time interval, the offset $\Delta V_S = -0.23$ V is added up to the nominal value $V_S$ of the DC voltage source in oscillator 1. As may be evinced by inspecting Figure 11, the relative phase of cell 1 undergoes a sudden shift by approximately $180°$, and, thereafter, the phase dynamics of the network evolve toward a new stable steady-state pattern, whereby $G(\boldsymbol{\varphi})$ is found to sit at its global minimum level, particularly $-6$ (see plot (b))), and the network is able to identify the chromatic number of the graph in Figure 10(a), as determined through the proposed iterative vertex coloring procedure (refer to plot (c)).

*5.3. Discussion*

Since a single crossover or pulse destabilisation manoeuvre may be unable to pull a more complex memristive network out of an impasse situation, or to reach the solution associated to the global minimum of the optimisation goal function, in case the dynamical system receives enough energy to move out of the solution associated to a local minimum of the optimisation goal function[25], a good approach to address this issue would be to reapply either of the two proposed strategies periodically, interchanging the connections of two distinct appropriate oscillators or applying a suitable destabilising pulse to a different ad-hoc oscillator at regular time intervals[26].

Let us provide a proof of principle of the proposed approach, focusing on the pulse destabilisation control strategy. Similar results were obtained through the periodic application of the crossover paradigm. With reference to Figure 13, plot (a) shows the time evolution of the phase shifts of the oscillators with labels running from 1 to 24 with respect to the reference cell 0 in a capacitively-coupled network of $NbO_x$ memristor oscillators implementing a $N = 25$-node undirected graph known as queen5_5 [50]—see also the inset in plot (b)—in case compensation for the mismatch between the capacitive loads of the oscillators and for the memristor device-to-device variability is set in place, and for the case where, periodically, on the basis of the two-step strategy introduced in Section 5.2, a distinct oscillator is perturbed by means of a $T_p$-long pulse of appropriate height $\Delta V_s$. As shown in plot (b) the optimisation goal function evolves progressively through various local minima before attaining the global minimum value, which is associated to the identification of the chromatic number of the queen5_5 graph, i.e., $n = 5$, as demonstrated in plot (c). After a 38 ms-long transient time interval, the network exhibits a robust oscillatory solution, given that the subsequent application of destabilisation pulse stimuli to the oscillators does no longer affect the phase dynamics.

---

[25] In some cases, after overcoming the impasse situation, the dynamical system could approach a new oscillatory solution associated to another local minimum of $G(\boldsymbol{\varphi})$.

[26] The time separation $T_{int}$ between consecutive applications of the crossover or pulse destabilisation strategy is set to 2 ms in the simulations discussed in this section. Furthermore, in this work the estimation of the common period $T$ of the oscillations appearing in a $N$-cell network, and the associated group ordering of the phase shifts of the cells $1, \ldots, N-1$ relative to the null phase of the reference cell 0 are carried out every cycle throughout the duration of any simulation. Moreover, in the first (latter) control strategy, the application of a pulse to the same oscillator (a crossover involving either oscillator from the same pair) is not allowed until at least 5 iterations of the control strategy have elapsed first. As a result, each pulse destabilisation (crossover) manoeuvre targets a different oscillator (involves a different pair of oscillators).

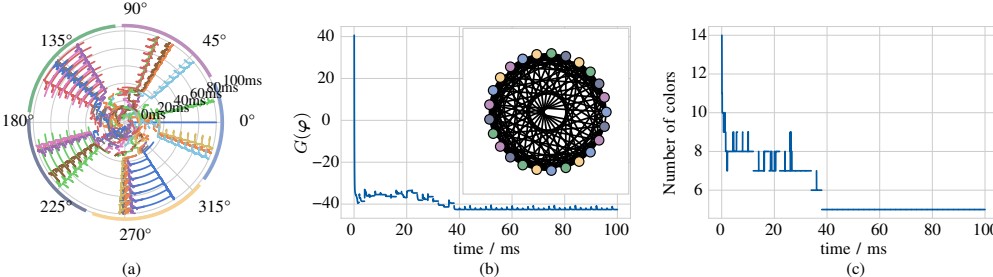

**Figure 13.** Evidence for the capability of a 25-cell network, preliminarily compensated for the unbalance in the number of connections per oscillator, and for the inter-device variability inherent to memristors, to converge toward the optimal solution of the vertex coloring problem for the graph queen5_5 [50]. The cyclic application of a pulse stimulus of fixed length and appropriate amplitude to an ad-hoc oscillator of the network guides the phase dynamics toward the global minimum solution. (**a**) Phase diagram visualising the time evolution of the phase of each oscillator $j \in \{1, \ldots, 24\}$ relative to the phase of the reference oscillator 0. (**b**) Time waveform of the optimisation goal function $G(\boldsymbol{\varphi})$. (**c**) Progression of the outcome of the iterative vertex coloring procedure of Section 4 over time. Throughout the second half of the simulation the minimum number of color groups, assigned to the vertices of the graph queen5_5, visualised in the inset of plot (**b**), is fixed to the chromatic number $n = 5$ of the graph itself, despite the network is subject to further pulse-based perturbations.

Table 3 shows a comparison between the solutions of the node coloring task for a number of graphs [50] pertaining to the $2^{nd}$ algorithm implementation challenge for NP-hard problems in Discrete Mathematics and Theoretical Computer Science (DIMACS) [51], derived from the application of various techniques, namely an algorithmic approach known as Brélaz heuristic [52], methods based upon the analysis of the phase dynamics of capacitively-coupled arrays of locally-active memristor oscillators without a control strategy for bypassing local minima solutions, as respectively presented in [42], and in Section 4, and, finally, paradigms including either a reconfigurability of the oscillators' couplings, as discussed in Section 5.1, or a perturbation of the memristive array, as presented in Section 5.2, to enable the dynamical system to exit an impasse state, and to resume the calculations of the problem solution, thereafter.

The results obtained through the analysis of the phase dynamics in memristive oscillatory networks—refer to the approach employed in [42] and to our iterative strategy from Section 4, where no technique to bypass local minima solutions is set in place, are comparable to the corresponding ones of the Brélaz heuristic algorithm. As may be evinced from Table 3, the graph coloring paradigm implemented through capacitively-coupled memristive oscillators in [42] classifies the vertices of all investigated graphs, with the exception of the one called queen6_6 [50], into a number of color groups equal to or lower than the number of colors assigned to the nodes of the corresponding graphs through the proposed iterative strategy from Section 4. However, it should be pointed out that, while the nominal parameter setting in cell and coupling circuits is unaltered in the numerical investigations of the networks of all the graphs in Table 3, it is unclear whether the same values were assigned to the physical attributes of the components of the array for the simulations of the corresponding systems in [42]. Furthermore, while the mathematical characterisation of our $NbO_x$ resistance switching memory is rooted on strong physics foundations, and is endowed with device-to-device variability control, the simplistic, model adopted to characterise the locally active $VO_2$ memristor in [42], has no physics basis, assuming that the two-terminal element may feature at any given time one of two conductance values, denoting the metallic and insulating state, respectively, depending upon the voltage falling across it, and does not account for the inherent spread in the device static and dynamic properties from sample to sample. The application of our iterative graph coloring strategy to the vertex orderings derived from the simulations of the networks from Table 3, for the case where the tendency of the oscillators' phase shifts to approach local

minima solutions is counterbalanced through the implementation of either the crossover or the pulse destabilisation control paradigms, leads to an evident performance improvement. With reference to each of the graphs—namely mycie15, queen5_5, queen6_6, queen7_7, and queen8_8—whereby, according to the iterative strategy from Section 4, our memristive network is unable to identify the chromatic number $n$ on its own, the periodic application of either of the two control paradigms from Sections 5.1 and 5.2 allows the lowest number of colors assigned to the $N$ vertices to decrease, and, in most cases, the final phase pattern of the memristive oscillatory array allows to determine the global minimum solution. As an example, which also reveals how further studies are necessary to improve the performance of our memristive networks in coloring the vertices of complex graphs, Figure 14(a) shows the phase dynamics of a balanced network implementing the queen6_6 graph [50], for the case where the mismatch in the number of couplings per oscillator and the memristor device-to-device variability are respectively compensated via the methodologies described in Sections 3.2 and 3.3, and a periodic application of the crossover control paradigm of Section 5.1, involving a distinct pair of oscillators from cycle to cycle, is set in place. In this case, the network keeps in a transient phase throughout the simulation. As may be evinced by inspecting the time evolution of the optimisation goal function $G(\varphi)$, shown in plot (b), and the evolution of the outcome of our iterative vertex coloring strategy over time, illustrated in plot (c), the dynamical system does not exhibit a monotonic decrease toward the global minimum solution, escaping the best solution, computed around 50 ms, to approach higher local minima thereafter. With regard to our intention to enhance the local minima bypass paradigms further, the analysis of the potentially-beneficial impact of the memristor thermal noise source on the capability of the dynamical system to descend monotonically toward the global minimum solution is one of the future research activities in our agenda.

**Remark 4.** *The first priority in our research agenda is to realize a hardware prototype able to solve various graph coloring problems of small/medium size on the basis of the oscillators' phase dynamics. In order to allow a hardware implementation of the proposed memristive computing engine to solve a number of different graph coloring problems, the connections between the processing units need to be adjustable on a case by case basis, which calls for the use of a coupling arrangement typical of a Hopfield neural network, with the introduction of a transistor switch, controllable via some ad hoc multiplexer, in series with each capacitor $C_C$. While hardware architecture considerations for large networks are still quite premature, we believe that scaling up the size of the computing engine would require the use of an array-like structure, as typically used for memories, to implement the programmable coupling circuitry. Except for the memristors, which could be arranged in crossbar configuration across the metal layers, the rest of the circuitry, necessary to implement the oscillatory cells, would be laid out on the CMOS substrate. In later generations of the proposed hardware, also back-end of line (BEOL) transistors can be envisioned so as to further increase the area efficiency of the computing platform. From a problem-solving perspective, scaling up the network size to tackle problems of bigger dimension, envisaging, in general, a larger number of connections between the vertices of the associated graphs, shall result in an inevitable increase in the number of local minima for the optimization goal function, which complicates the operation of the control circuitry, as it tries to guide the oscillators' phases toward the optimal grouping at steady state. This is a general problem for all state-of-the-art software algorithms and hardware platforms, which aim to minimize non-convex optimization goal functions. In order to solve graph coloring problems of higher complexity, some fine tuning of the control strategies, proposed in this manuscript, as inspired by the most efficient NP-hard optimization problem solvers, available today, is expected to be necessary.*

**Table 3.** Comparison between the solutions of the vertex coloring problem for various graphs [50] for the 2$^{nd}$ algorithm implementation challenge for NP-hard problems in DIMACS [51], obtained through the application of a specific algorithm, known as Brélaz heuristic [52], by means of methods exploiting the phase dynamics of capacitively-coupled memristive networks without a control strategy for bypassing local minima solutions, namely the technique proposed in [42], and the iterative node coloring procedure, presented in Section 4, and via the iterative node coloring procedure augmented with strategies, based upon crossover and pulse destabilisation, presented in Sections 5.1 and 5.2, respectively, and aimed to overcome local minima solutions [31]. The results tabulated in the last three columns were computed through the analysis of 100 ms long numerical simulations.

| Minimum Number of Color Groups for the Classification of the Vertices of the Associated Group | | | | | | | |
|---|---|---|---|---|---|---|---|
| graph | vertices | $n$ | Brélaz algorithm | [42] | iterative strategy | iterative strategy and crossover control | iterative strategy and pulse destabilisation control |
| mycie13 | 11 | 4 | 4 | 4 | 4 | 4 | 4 |
| mycie14 | 20 | 5 | 5 | 5 | 5 | 5 | 5 |
| mycie15 | 47 | 6 | 6 | 6 | 7 | 6 | 6 |
| queen5_5 | 25 | 5 | 7 | 6 | 7 | 5 | 5 |
| queen6_6 | 36 | 7 | 10 | 12 | 11 | 8 | 8 |
| queen7_7 | 49 | 7 | 12 | 12 | 14 | 10 | 10 |
| queen8_8 | 64 | 9 | 15 | 14 | 15 | 13 | 13 |

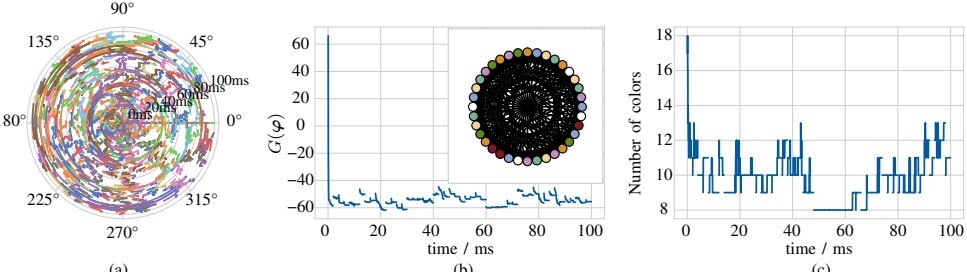

(a)  (b)  (c)

**Figure 14.** (**a**) Phase dynamics of the network associated to the graph queen6_6 [50], after its preliminary compensation for the unbalance in the number of connections per oscillator, and for the inter-device variability inherent to memristors, upon the periodic interchange between the couplings of two specific oscillators. In this case the phase dynamics of the network keep in a transient state throughout the 100 ms-long simulation. (**b**) Evolution of the optimisation goal function $G(\varphi)$ over time. (**c**) Minimum number of colors, assigned to the nodes of the graph queen6_6 through the iterative vertex coloring procedure of Section 4, applied once every cycle, versus time. Half way through the simulation the nodes of the graph, illustrated in the inset of plot (**b**), are classified into 8 color groups, one more than the correct number (refer to Table 3), but this solution proves to be unstable, when the network, thereafter, is subject to further crossover-based perturbations.

## 6. Conclusions

The local activity [26] of NbO$_x$ memristors ([27,28]) allows the emulation of neuronal dynamics ([9,11]), the implementation of bio-inspired signal processing paradigms [53], and the reproduction of complex phenomena [30] emerging in systems from cellular biology [44]. This manuscript serves as a pedagogical tutorial to the operating principles of a cellular nonlinear network of oscillators, coupled through linear capacitors, and employing one locally-active memristor [43] each, recently introduced in [31] to solve a non-deterministic polynomial (NP)-hard combinatorial optimization problem, known as vertex coloring. While, due to page limitation, only a compact description of the signal processing paradigm, implemented by the proposed Memristor Oscillatory Network, was reported in [31], this tutorial reports all the details of the mechanisms underlying its *modus operandi*. Importantly, control methods [33] to compensate for the inherent variability of memristor devices, to counteract the imbalance between the load capacitances of the oscillators, as well as, most importantly, to prevent the bio-inspired network to attain a sub-optimal steady state, are developed and implemented in circuit form. The Memristor Oscillatory Network, endowed with the proposed control circuitry, is found to outperform state-of-the-art software and hardware competitor alternatives, identifying, for



each graph from a wide selection, the lowest number of color groups for the respective vertices. As a more general conclusion, the potential of all locally-active devices, including niobium ([28,43,54]) or vanadium dioxide [55] threshold switches, and ovonic threshold switches [56,57], is expected to be subject to a thorough exploration, in the years to come, for a possible exploitation of their small-signal amplification capability for electronics applications, e.g. to build nano-oscillators with tuneable frequency ([58]), to solve NP-hard combinatorial optimization problems, as discussed in this manuscript, for reproducing complex biological phenomena [44], for exploring new forms of computing via pattern formation dynamics [59], or for designing bio-plausible neuromorphic circuits [10].

**Author Contributions:** All authors have read and agreed to the published version of the manuscript.

**Funding:** This research received no external funding.

**Data Availability Statement:** Not applicable.

**Conflicts of Interest:** The authors declare no conflict of interest.

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
