# Peer review of "Graph Coloring via Locally-Active Memristor Oscillatory Networks"

_jlpea, doi:10.3390/jlpea12020022_

Round 1

Reviewer 1 Report

The paper is devoted to actual problem of application of memristor based oscillatory networks.

The paper presents an approach to solve graph coloring problem using oscillatory networks of such type.

This combinatorial optimization problem is characterized as NP-hard problem.

The paper demonstrates the advantages of the developed approach to solve high complexity problems of this type. The iterative procedure using network phase dynamics is described.

Authors claim that this paper can be considered as a pedagogical tutorial. The paper fully satisfies to requirements of pedagogical tutorial within the frames of the considered problem.

The authors are well known experts in memristor circuitry.

The paper is of interest to specialists.

The paper can be recommended for publication.

Author Response

Please see the attachment, thanks

Reviewer 2 Report

The manuscript by Ascoli et al. solves the graph coloring problem by leveraging the NbOx locally active memristor oscillatory network, featuring improved strategies for non-uniform coupling structure and methods to bypass the local minima. The manuscript is systematically organized with well-prepared plots. Some technical questions are as following.

  1. The references [31] and [39] are the same. Please double check.
  2. The statement “…capacitive impedance ZCi (jw), loading oscillator i for each value of i in the set {0, 1, 2} …”, however, ZC1 (jw), ZC2 (jw), ZC3 (jw) are defined in Eqs. (4)-(6). Please double check and unify: {0, 1, 2} or {1, 2, 3} since it’s a bit confusing. Also, please explain the meaning of C in Eq. (7).
  3. I am wondering if there is a theoretical method to adjust RS of oscillators? After changing the resistance RS of different oscillators, whether the frequencies of every single oscillator are the same? What is the suboptimal (the optimal) initialization setting? (Does it mean that the obtained solution is suboptimal or optimal under these initial conditions, if so, how to identify the suboptimal (the optimal) initialization setting?)
  4. Could the authors discuss in paper if other types of threshold switches, such as ovonic chalcogenide devices or diffusive memristors, are locally active and thus bearing the same application?

Reviewer 3 Report

The work by Alon Ascoli et al. reports on a theoretical and simulation analysis of the memristive oscillatory networks for the graph coloring problem. The oscillators in the network are built upon threshold switching memristive device based on NbOx and are coupled by capacitors. The graph coloring problem was solved by the automatically phase clustering of the phases of the oscillators. The simulation and analysis are based on the physical based compact model of NbOx memristors. The work is interesting, and the presented analysis and discussion are solid. I recommend the paper to be accepted after solving several minor issues:

  1. How is the coupled-oscillator network related to the Cellular Nonlinear Network? Cellular Nonlinear Network has different structure and operational mechanism (L. Chua, 1988) when compared with the oscillator network structure in this work.
  2. The oscillators are all coupled by capacitors. Does the capacitance of the capacitors matter? Would the network capable of solving problems that need tunable coupling strength?
  3. The memristor device-to-device variability seems a big issue. Do the oscillators need to have same intrinsic oscillation frequency for working?
  4. Could the authors comment on the how to scaling up the network size to tackle graph problems of bigger size?
